

# *Sphenofontis velserae* gen. et sp. nov., a new rhynchocephalian from the Late Jurassic of Brunn (Solnhofen Archipelago, southern Germany)

Andrea Villa[1,2], Roel Montie[2], Martin Röper[2,3], Monika Rothgaenger[2,3] and Oliver W.M. Rauhut[2,4,5]

[1] SNSB—Bayerische Staatssammlung für Paläontologie und Geologie, Munich, Germany
[2] Dipartimento di Scienze della Terra, Università degli Studi di Torino, Torino, Italy
[3] Museum Solnhofen, Solnhofen, Germany
[4] Department of Earth and Environmental Sciences, Ludwig-Maximilians-Universität, Munich, Germany
[5] GeoBioCenter, Ludwig-Maximilians-Universität, Munich, Germany

Corresponding author
Andrea Villa, a.villa@unito.it

## ABSTRACT

The Solnhofen Archipelago is well known for its fossil vertebrates of Late Jurassic age, among which figure numerous rhynchocephalian specimens, representing at least six and up to nine genera. A new taxon, named *Sphenofontis velserae* gen. et sp. nov., increases rhynchocephalian diversity in the Solnhofen Archipelago and is herein described based on a single, well-preserved specimen originating from the Late Kimmeridgian of the Brunn quarry, near Regensburg. The exquisite preservation of the holotype allowed a detailed description of the animal, revealing a skeletal morphology that includes both plesiomorphic and derived features within rhynchocephalians. *Sphenofontis* is herein referred to Neosphenodontia and tentatively to sphenodontine sphenodontids. It notably differs from all other rhynchocephalians known from the Jurassic of Europe, showing instead closer resemblance with the Middle Jurassic *Cynosphenodon* from Mexico and especially the extant *Sphenodon*. This is evidence for a wide distribution of taxa related to the extant tuatara early in the Mesozoic, and also for the presence of less-specialized rhynchocephalians coexisting with more derived forms during the earliest time in the history of the Solnhofen Archipelago.

## INTRODUCTION

Fossils of rhynchocephalians from the Jurassic of the Solnhofen Archipelago (formerly often collectively called "Solnhofen limestones"; for an overview of the geology and history of nomenclature of geological units see *Niebuhr & Pürner, 2014*), in Germany, are known since at least the first half of the 19th century (*Goldfuss, 1831*; *Meyer, 1831*; *Fitzinger, 1837*; *Meyer, 1845*; *Meyer, 1847*), even though at least some of them were not recognised as such originally. By the current state of knowledge, the different units of limestones have to

date yielded at least six and up to nine different rhynchocephalian genera (*Cocude-Michel, 1963*; *Cocude-Michel, 1967a*; *Cocude-Michel, 1967b*; *Fabre, 1981*; *Rauhut et al., 2012*; *Tischlinger & Rauhut, 2015*; *Bever & Norell, 2017*). Among these, *Homoeosaurus* Meyer, 1947, *Oenosaurus Rauhut et al., 2012*, *Pleurosaurus Meyer, 1831*, and *Vadasaurus Bever & Norell, 2017* are all considered valid, without any controversy. Another, large-bodied rhynchocephalian was described under the name *Piocormus* by *Wagner (1852)*. This taxon, known from a single specimen from the Solnhofen Archipelago (see also *Cocude-Michel, 1967b*), is generally similar to *Sapheosaurus*, a common genus from the Kimmeridgian of Cerin, France (*Cocude-Michel, 1963*; *Fabre, 1981*), which also seems to occur in some localities of the Solnhofen Archipelago (*Tischlinger & Rauhut, 2015*). However, whereas *Evans (1994)* suggested that these genera might be synonymous, *Cocude-Michel (1963*, *1967b)* and *Fabre (1981)* considered them to be separate taxa. A further genus is represented by fossils formerly attributed to either *Kallimodon Cocude-Michel, 1963* or *Leptosaurus Fitzinger, 1837*. These two genera were synonymized by *Fabre (1981)*, with *Leptosaurus* having priority, but this synonymization was not unreservedly accepted by subsequent authors (e.g., *Rauhut & Röper, 2013*; *Rauhut & López-Arbarello, 2016*; *Rauhut et al., 2017*). Refuting this synonymization would increase the count of rhynchocephalian genera from the Solnhofen limestones to at least seven, but only further studies dealing with this issue will resolve this. In the context of this paper, we treat *Kallimodon* as a separate taxon from *Leptosaurus*. Finally, the genus name *Acrosaurus* has been coined for small aquatic rhynchocephalians from the Solnhofen Archipelago (*Meyer, 1854*). These small animals have repeatedly been argued to be juvenile specimens of *Pleurosaurus* (e.g., *Hoffstetter, 1955*; *Rothery, 2002*), but have been regarded as a valid further taxon of rhynchocephalians by others (e.g., *Cocude-Michel, 1963*).

Apart from these formally named taxa, a number of so far unnamed species are present in the Solnhofen Archipelago (*Tischlinger & Rauhut, 2015*). *Rauhut et al. (2017)* already pointed out the presence of a further taxon differing considerably from all other rhynchocephalians from the limestones. This taxon, represented by a single specimen coming from the site of Brunn, is part of a diverse vertebrate fauna, including chondrichthyans, osteichthyans, marine turtles, crocodyliforms, pterosaurs, as well as three other rhynchocephalian specimens. The scope of the present work is to describe this specimen in detail, define its taxonomic identity and phylogenetic affinities, and discuss some of its unique characteristics.

## Geological and Paleontological context

The Kimmeridgian-Tithonian laminated limestones of southern Germany have long been recognized for their abundant and especially exceptionally preserved fossils (see *Barthel, Swinburne & Conway Morris, 1990*; *Arratia et al., 2015*). Although these units have long collectively been known as the "Solnhofen limestones", recent geological and stratigraphic work has helped to differentiate separate units representing different local settings and stratigraphic horizons (see *Schweigert, 2007*, *2015*; *Niebuhr & Pürner, 2014*; *Viohl, 2015*). Therefore, the term "Solnhofen Archipelago" has recently been established for the regional context of these limestones (e.g., *Röper, 2005*; *López-Arbarello & Schröder, 2014*).

The locality of Brunn (Fig. 1) is placed in the most eastern and northern part of the area usually included in the Solnhofen Archipelago. It is found in the Upper Palatinate region, some 15 km north-west of the city of Regensburg. Geologically, the locality Brunn is placed at the southern rim of the small Pfraundorf-Heitzenhofener basin (*Röper, 1997*), in a series of intercalated massive and laminated limestones that can be assigned to the Ebenwies Member of the Torleite Formation. A total of eight different layers of plattenkalk are exposed in a complete outcropping section of c. eight metres of Late Jurassic sediments in the Brunn quarry (*Röper, Rothgaenger & Rothgaenger, 1996*; *Röper, 1997*; *Heyng, Rothgaenger & Röper, 2015*), with all of these layers having yielded vertebrate remains (*Rauhut et al., 2017*). The rhynchocephalian specimens known from the locality Brunn (*Rauhut & Röper, 2013*; *Rauhut et al., 2017*) were found in plattenkalk layer 2, a less than 50 cm thick layer of finely laminated limestone within the lowermost 2 m of the section.

The locality Brunn is notable for the abundance of fossil plants, which account for up to one-fourth of the macrofossils found (*Röper, Rothgaenger & Rothgaenger, 1996*; *Heyng, Rothgaenger & Röper, 2015*). Apart from a diverse marine invertebrate fauna, including clades to be expected in a Late Jurassic marine setting, the vertebrate fauna is dominated by abundant actinopterygians, including ginglymodians, halecomorphs, and abundant teleosts (*Rauhut et al., 2017*). Tetrapods are generally rare and include a few aquatic turtles, pterosaurs, an atoposaurid crocodylomorph, and rhynchocephalians (*Rauhut et al., 2017*).

## MATERIALS & METHODS

SNSB-BSPG 1993 XVIII 4 was described following the terminology proposed by *Evans (2008)* for the cranium, *Hoffstetter & Gasc (1969)* for the axial skeleton, and *Russell & Bauer (2008)* for the appendicular skeleton. Detailed photos of the jaws and the cervical region were taken with a Leica M165 FC microscope equipped with a DFC450 camera and the Leica Application Suite (LAS) 4.5. UV-light documentation followed the methodology described by *Tischlinger (2015)* and *Tischlinger & Arratia (2013)*.

The electronic version of this article in Portable Document Format (PDF) will represent a published work according to the International Commission on Zoological Nomenclature (ICZN), and hence the new names contained in the electronic version are effectively published under that Code from the electronic edition alone. This published work and the nomenclatural acts it contains have been registered in ZooBank, the online registration system for the ICZN. The ZooBank LSIDs (Life Science Identifiers) can be resolved and the associated information viewed through any standard web browser by appending the LSID to the prefix http://zoobank.org/. The LSID for this publication is: urn:lsid:zoobank.org:pub:177F78D8-2C99-4C3B-8ED5-8D8ADE960A57. The online version of this work is archived and available from the following digital repositories: PeerJ, PubMed Central and CLOCKSS.

## SYSTEMATIC PALEONTOLOGY

Lepidosauria *Haeckel, 1866*
Rhynchocephalia *Günther, 1867*
Sphenodontia *Williston, 1925*

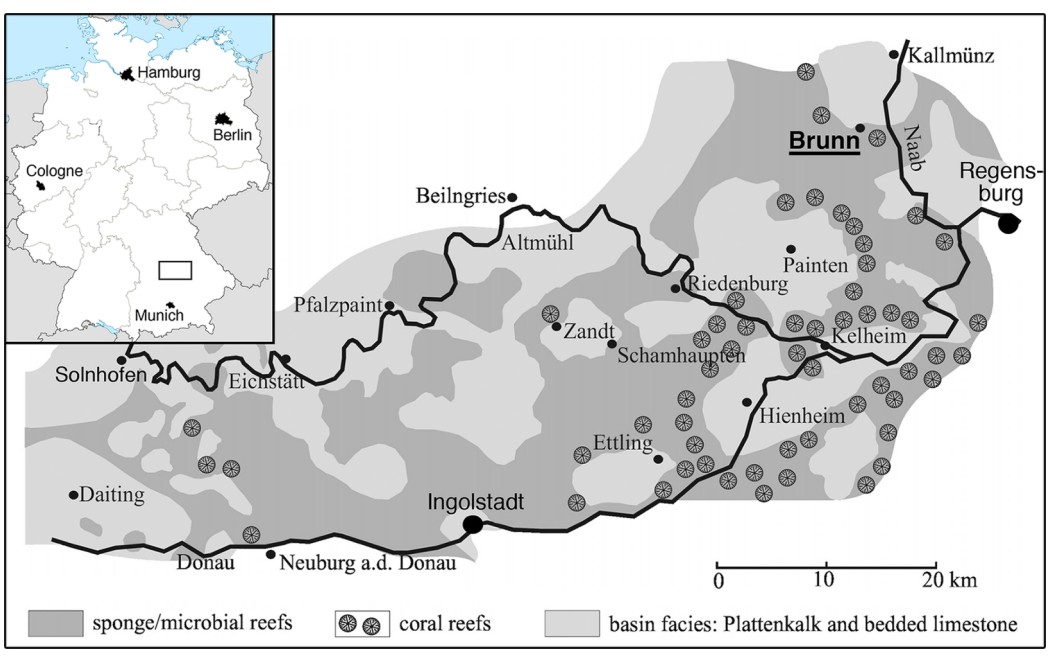

**Figure 1 Map of the area between Solnhofen and Regensburg.** The map shows the paleogeographic reconstruction of the Solnhofen Archipelago, as well as the current position of Brunn.

Eusphenodontia *Herrera-Flores et al., 2018*
Neosphenodontia *Herrera-Flores et al., 2018*
Sphenodontidae *Cope, 1871*
Sphenodontinae *Cope, 1871*
*Sphenofontis* gen. nov.
*Sphenofontis velserae* sp. nov.

**Holotype.** SNSB-BSPG 1993 XVIII 4, a slab containing a nearly complete and articulated skeleton (Fig. 2).

**Type locality and horizon.** "Plattenkalk layer 2" (*Rauhut & Röper, 2013*; *Rauhut et al., 2017*), Brunn quarry, Ebenwies Member, Torleite Formation, Bavaria, Germany; Late Kimmeridgian (Subeumela Subzone; *Röper & Rothgaenger, 1995*, *1997*; *Schweigert, 2007*; *Heyng, Rothgaenger & Röper, 2015*).

**Etymology.** Genus name combines the prefix *Spheno-*, with reference to the taxon being a sphenodontian, and the latin word *fontis*, genitive of *fons* (= spring, but also well), roughly meaning "the sphenodontian of the well". This acknowledges the origin of the name of the type locality Brunn, which comes from the German Brunnen (= well). Species name honours Lisa Velser, who discovered and prepared the holotype specimen.

**Diagnosis.** *Sphenofontis velserae* can be diagnosed by at least three possible autapomorphies: a medially-displaced fourth additional tooth in the maxilla; proximally-constricted and
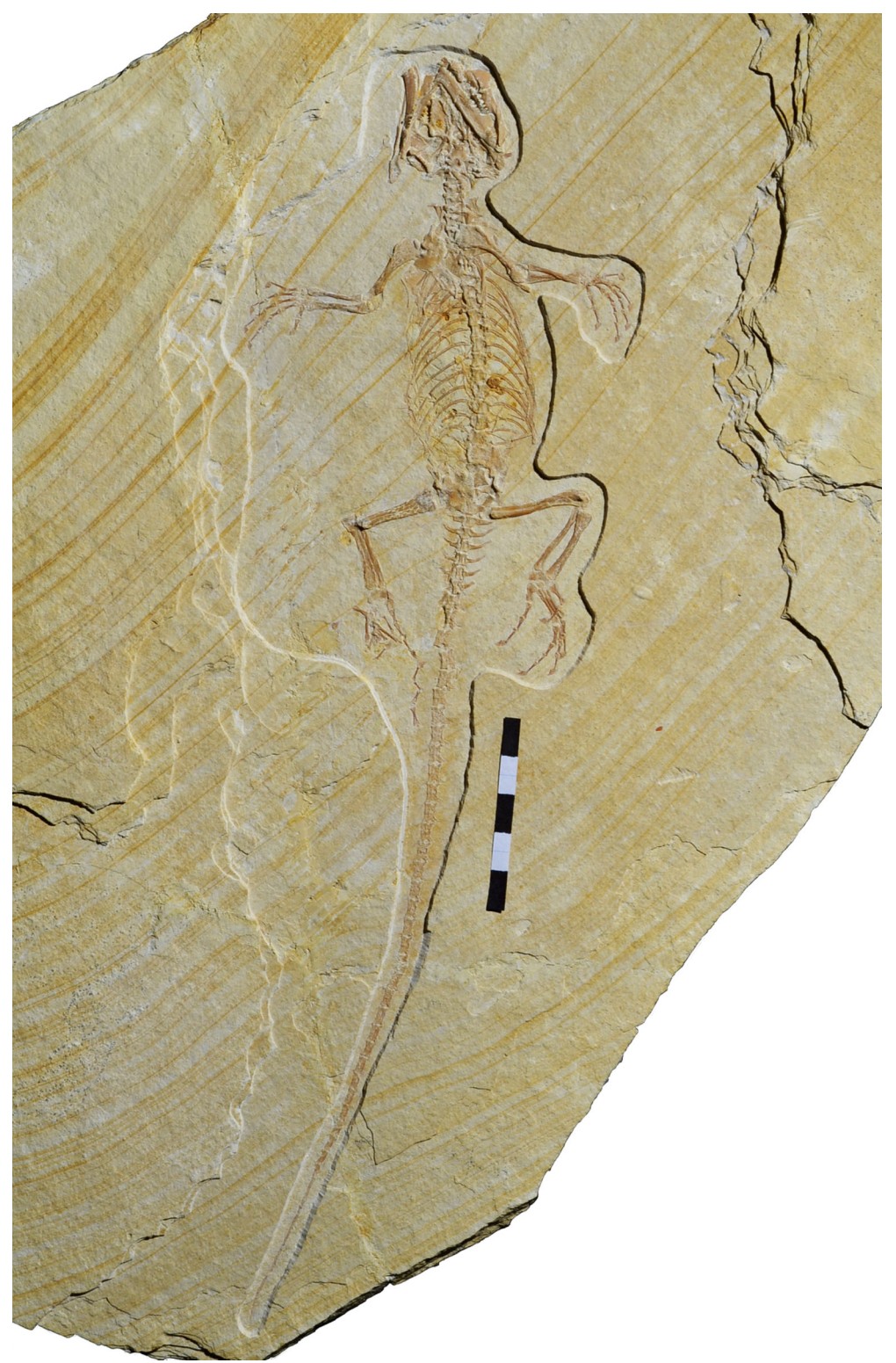

**Figure 2 Holotype of *Sphenofontis velserae* gen. et sp. nov., SNSB-BSPG 1993 XVIII 4.** Each subdivision of the scale bar is 1 cm.

strongly distally-expanded transverse processes of the first sacral vertebra; and anterolaterally-oriented transverse processes of the first caudal vertebra.

Apart from these, *S. velserae* differs from non-Neosphenodontian sphenodontians in having only one row of palatine teeth and no pterygoid teeth. It also differs from *Diphydontosaurus*, *Whitakersaurus*, and *Deltadectes* in the completely acrodont dentition; from *Gephyrosaurus* in the open Meckelian fossa and the acrodont dentition; from *Sphenocondor* in the long ventral posterior process of the dentary; from both *Pelecymala* and *Sigmala* in the wider facial process and the presence of successional teeth; and from *Tingitana* in the presence of an enlarged mandibular foramen. It differs from clevosaurs in the well-developed anterior premaxillary process of the maxilla, the anterior process of the jugal not reaching the anterior end of the orbit, and the postorbital ending dorsomedially in a lateral notch of the postfrontal. It further differs from *Clevosaurus bairdi* in the lower facial process; from *Clevosaurus brasiliensis* in having short and thick basipterygoid processes; from *Clevosaurus minor* in having palatine teeth supported by a ridge; and from *Clevosaurus hudsoni* in the presence of a coronoid bone and the lower count of postpelvic intercentra.

*Sphenofontis velserae* differs from *Homoeosaurus*, *Kallimodon*, *Leptosaurus*, and pleurosaurids in having a regionalized dentition. It differs from *Homoeosaurus maximiliani*, *Kallimodon*, and *Leptosaurus* in the higher count of presacral vertebrae and in retaining intercentra in the dorsal region of the column; from *Homoeosaurus solnhofensis* in the longer tail and in retaining intercentra in the dorsal region of the column; and further from *Kallimodon* and *Leptosaurus* in tail autotomy starting more anteriorly in the tail. It differs from *Oenosaurus* in the absence of a strongly-developed medial process of the maxilla, in the anterior process of the jugal not reaching the anterior end of the orbit, in the absence of a triangular lateral end of the ectopterygoid, in the flat ventral surface of the sphenoid, in the steeply-inclined mandibular symphysis, in the coronoid process of the dentary lower than the depth of dentary itself anterior to the process, in the lower coronoid bone, and in the dentition not composed by tooth plates. It differs from *Piocormus* and *Sapheosaurus* in the presence of discrete marginal teeth, the higher count of presacral vertebrae, and the lower count of postpelvic intercentra. It differs from pleurosaurids in the shorter skull, the shorter retroarticular process of the lower jaw, the lower count of presacral vertebrae, and in the metacarpal 1 not enlarged proximally. *Sphenofontis* further differs from *Vadasaurus* in the presence of a quadratojugal process, the postorbital ending dorsomedially in a lateral notch of the postfrontal, and in retaining intercentra in the dorsal region of the column; and from *Pleurosaurus* in the retention of tail autotomy.

*Sphenofontis velserae* differs from *Pamizinsaurus* in the steeply-inclined mandibular symphysis and the absence of osteoderms; from *Opisthias* in the higher number of successional dentary teeth; from *Theretairus* in the lower number of caniniform teeth; and from *Ankylosphenodon* in retaining intercentra in the dorsal region of the column, in a less robust first digit, and in not having a continuously-growing and unregionalized dentition. It differs from all Eilenodontinae in the slender lower jaw and in not having an opisthodontian dentition. It also differs from *Priosphenodon* in the well-developed anterior

**Table 1  Relevant measurements of the axial skeleton of SNSB-BSPG 1993 XVIII 4.**

| | |
|---|---:|
| **Cranium** | |
| *Cranial length* | 28* |
| *Cranial width* | 27* |
| *Maxilla, length* | (14) |
| **Lower jaw** | |
| *Lower jaw, length* | 30 |
| *Dentary, length* | 25 |
| **Vertebral column** | |
| *Length of presacral region* | (79) |
| *Length of sacral region* | (6.5) |
| *Sacral vertebra 1, proximal width of transverse process* | 3.3 |
| *Sacral vertebra 1, minimal width of transverse process* | 0.6 |
| *Sacral vertebra 1, distal width of transverse process* | 3.1 |
| *Sacral vertebra 1, length of transverse process* | 5.2 |
| *Sacral vertebra 2, proximal width of transverse process* | 3.05 |
| *Sacral vertebra 2, distal width of transverse process* | 2.25 |
| *Sacral vertebra 2, length of transverse process* | 5.5 |
| *Tail length* | 221 |

**Note:**
All measurements are expressed in mm. Asterisks mark measurements estimated based on poorly preserved elements, whereas parentheses represent those referred to skeletal portions that are not complete (or cannot be confidently measured in their completeness) in SNSB-BSPG 1993 XVIII 4.

premaxillary process, the narrower facial process, the anterior process of the jugal not reaching anteriorly the end of the orbit, and the sinusoid ventral margin of the dentary; and from *Priosphenodon avelasi* in particular in not having squared distal phalanges. It differs from *Sphenotitan* in the well-developed anterior premaxillary process, the higher facial process, and in having only one row of palatine teeth and no pterygoid teeth. It differs from *Sphenovipera* in the longer posterior portion of the lower jaw, wide adductor fossa, lower number of caniniform teeth, and absence of a dorsoventral groove on the latter; from *Cynosphenodon* in the lower facial process and the steeply-inclined mandibular symphysis; and from *Kawasphenodon* in the dentary teeth not limited to the posterior portion of the tooth row, teeth not grooved posteriorly, and, at least from *Kawasphenodon expectatus*, in the sinusoid ventral margin of the dentary. It differs from *Sphenodon* in the presence of the posterodorsal process of the premaxilla, the wider and lower facial process of the maxilla, the articulation between postorbital and postfrontal visible in ventral view, the flat ventral surface of the sphenoid, the forked transverse process of the second sacral vertebra, the first autotomic caudal vertebra located more anteriorly in the tail, and the lower count of postpelvic intercentra.

**Table 2 Relevant measurements of the appendicular skeleton of SNSB-BSPG 1993 XVIII 4.**

| | | | |
|---|---|---|---|
| **Pectoral girdle** | | Pubis, mediolateral width | 10.4 |
| Interclavicle, anterior width | 8.3 | Pubis, maximum width of medial process | 5.15 |
| Interclavicle, length | (10) | Pubis, minimum width of medial process | 4.5 |
| Scapulocoracoid, length | L: 13.8 | Pubis, maximum length | 8.1 |
| Coracoid, width | L: 0.5; R: 0.59 | Pubis, width from medial (distal) end to tubercle | 6.7 |
| **Forelimb** | | Pubis, width from tubercle to lateral (proximal) end | 3.7 |
| Forelimb, length | 54.5 | Pubis, length from pectineal tubercle to midline | 3 |
| Humerus, length | L: 21.7; R: 21.2 | Pubis, length from midline to obturator foramen | 2 |
| Humerus, proximal epiphysis width | L: 0.7; R: 0.63 | Pubis, length from midline to ischium facet | 5 |
| Humerus, diaphysis width | L: 2; R: 2 | Ischium, maximum mediolateral width | 8.8 |
| Humerus, distal epiphysis width | L: 5.1; R: 5.6 | Ischium, maximum anteroposterior length | 8.6 |
| Radius, length | L: (13.6); R: 14.35 | Ischium, length of distal end | 7.8 |
| Radius, diaphysis width | 0.1 | Ischium, length of proximal end | 4.8 |
| Ulna, length | L: (13.6); R: 16.55 | **Hindlimb** | |
| Ulna, diaphysis width | 1.6 | Hindlimb, length | 76 |
| Manus, length | (22.6) | Femur, length | 28.5 |
| Carpus, width | R: 4.7 | Femur, length proximal to greater trochanter | 3.9 |
| Carpus, length | L: 2.9*; R: 2 | Femur, diaphysis width | 2.5 |
| Metacarpal 1, length | L: 4.3; R: 3.15 | Tibia, length | 20.05 |
| Metacarpal 2, length | R: 5.15 | Tibia, diaphysis width | 1.95 |
| Metacarpal 3, length | R: 6.05 | Fibula, length | 20.05 |
| Metacarpal 4, length | R: 5.05 | Fibula, diaphysis width | 1.4 |
| Metacarpal 5, length | R: 3.7 | Pes, length | 32.95 |
| Digit I, length of first phalanx | L: 4.7; R: 4.4 | Astragalocalcaneum, width | 5.7 |
| Digit II, length of first phalanx | L: 3.3; R: 3.2 | Astragalocalcaneum, length | 3 |
| Digit III, length of first phalanx | L: 3.3; R: 3.2 | Metatarsal 1, length | 5.9 |
| Digit IV, length of first phalanx | L: 3.25; R: 3.15 | Metatarsal 2, length | 8.4 |
| Digit V, length of first phalanx | L: 3.15; R: 3.60 | Metatarsal 3, length | 9.55 |
| Digit II, length of second phalanx | L: 4.3; R: 4.5* | Metatarsal 4, length | 10.3 |
| Digit III, length of second phalanx | L: 2.8; R: 3.1 | Metatarsal 5, length | 4.8 |
| Digit IV, length of second phalanx | L: 2.5; R: 3 | Metatarsal 5, width | 3.05 |
| Digit V, length of second phalanx | L: 4.5 | Digit I, length of first phalanx | 5.1 |
| Digit III, length of third phalanx | L: 3.9; R: 3.85 | Digit II, length of first phalanx | 4.05 |
| Digit IV, length of third phalanx | L: 2.75; R: 2.75 | Digit III, length of first phalanx | 4.7 |
| Digit IV, length of fourth phalanx | L: 3.85; R: 3.65 | Digit IV, length of first phalanx | 5.2 |
| Digit I, length of distal phalanx | L: 2.15; R: 2.5 | Digit V, length of first phalanx | 4.3 |
| Digit II, length of distal phalanx | L: 2.15; R: 2.2 | Digit II, length of second phalanx | 4.75 |

| | | | |
|---|---|---|---|
| *Digit III, length of distal phalanx* | L: 2.4; R: 2.2 | *Digit III, length of second phalanx* | 3.5 |
| *Digit IV, length of distal phalanx* | L: 2.2; R: 2.1 | *Digit IV, length of second phalanx* | 3.75 |
| *Digit V, length of distal phalanx* | L: 2.45 | *Digit V, length of second phalanx* | 3.85 |
| **Pelvic girdle** | | *Digit III, length of third phalanx* | 4.3 |
| *Ilium, total length* | (11.75) | *Digit IV, length of third phalanx* | 2.9 |
| *Ilium, length of posterior process* | 5.05 | *Digit V, length of third phalanx* | 4.55 |
| *Ilium, length of ischium facet* | 3.9 | *Digit IV, length of fourth phalanx* | 3.9 |
| *Ilium, length anterior to ischium facet* | 5* | *Digit I, length of distal phalanx* | 2.85 |

**Note:**

All measurements are expressed in mm. Asterisks mark measurements estimated based on poorly preserved elements, whereas parentheses represent those referred to skeletal portions that are not complete (or cannot be confidently measured in their completeness) in SNSB-BSPG 1993 XVIII 4. Measurements for left (L) and right (R) elements are reported when possible for paired bones.

### Description and comparisons

SNSB-BSPG 1993 XVIII 4 (Fig. 2) is practically complete and well preserved, but strongly flattened, as it is typical for fossils from laminated limestones. Due to this flattening, the skull is crushed and partially disarticulated. Furthermore, the right pes is disarticulated, with the fourth digit having been moved under the tail. The skeleton is exposed in ventral view. Relevant measurements are reported in Tables 1 and 2.

**Skull.** The skull (Fig. 3) is short and wide, almost as wide as it is long (maximally 27 mm wide and 28 mm long from the tip of the premaxilla to the occipital condyle, although the width may be slightly exaggerated by crushing). It has a subtriangular shape and a stocky aspect, much more like *Homoeosaurus* and maybe *Oenosaurus* and clevosaurids than the extant *Sphenodon* and fossil taxa with a more elongated skull, such as *Kallimodon*, *Leptosaurus*, *Piocormus*, *Sapheosaurus*, and especially pleurosaurids. The slight disarticulation of the elements of the snout hinders a completely confident recognition of the anterior profile of the skull, but it appears rather rounded. Most of the skull roof bones are not exposed, even though they are most likely still preserved (parts of the covered elements, including the frontal and the parietal, are visible through the palate bones). As in many sphenodontians, the orbit was very large: even though the full extent cannot be really seen, its anteroposterior length can be estimated as 12.5 mm. The lateral temporal fenestra was obviously considerably smaller; although its margins are not completely preserved on either side, its maximum anteroposterior length can be estimated to be no more than 9 mm, and the opening was probably rather in the range of 5–7 mm (based on the distance between the posterior margin of the ascending process of the jugal and the occipital condyle).

Most of the bones of the skull roof are either not preserved or covered by other elements, mainly of the palate. Parts of the frontals are visible in ventral view (Fig. 3). They seem to be fused without visible suture. They are constricted between the orbits and widen anteriorly towards the contact with the prefrontal. The orbital margins are notably swollen in ventral view, as in *Sphenodon* (*Jones et al., 2011*). The space between these

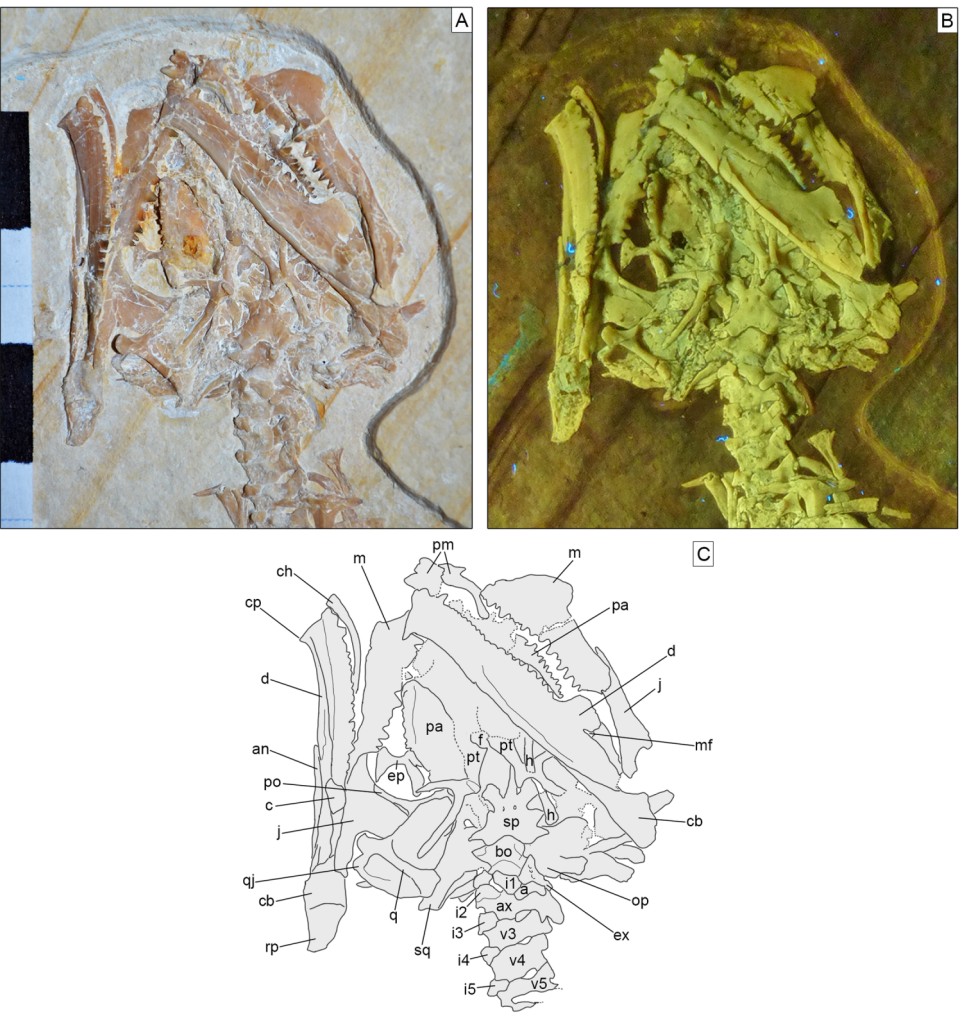

**Figure 3** Skull of *Sphenofontis velserae* gen. et sp. nov. (A) Standard light; (B) UV-light; (C) interpretative drawing. Each subdivision of the scale bar in A is 1 cm. Abbreviations: a, atlas; an, angular; ax, axis; bo, basioccipital; c, coronoid; cb, compound bone; ch, possible ceratohyal; cp, "chin" projection of dentary; d, dentary; ep, ectopterygoid; ex, exoccipital; f, frontal; h, possible element of the hyobranchial apparatus; i1-5, first to fifth intercentra; j, jugal; m, maxilla; mf, mandibular foramen; op, opisthotic; pa, palatine; pm, premaxilla; po, postorbital; pt, pterygoid; q, quadrate; qj, quadratojugal; rp, retroarticular process; sp, sphenoid; sq, squamosal; v3-5, third to fifth vertebrae. The UV-light photo in B was taken by Helmut Tischlinger.

swollen margins widens posteriorly to form the facets for the olfactory bulbs. The parietals are hidden by the ventral elements of the braincase.

The paired premaxillae (Figs. 3, 4, 5) are small, with the premaxillary body below the nares being considerably longer (2.5 mm) than high (c. 1.1 mm), as in *Planocephalosaurus* (*Fraser, 1982*) and *Sphenotitan* (*Martínez et al., 2013*), but in contrast to the short and high premaxillae in *Sphenodon* (*Jones et al., 2011*), *Priosphenodon* (*Apesteguía & Novas, 2003*; *Apesteguía & Carballido, 2014*), and *Clevosaurus* (*Fraser, 1988*; *Sues, Shubin & Olsen, 1994*; *Hsiou, De França & Ferigolo, 2015*). They have a small alveolar portion carrying three teeth on its ventral margin (Figs. 3, 4A, 5A). The medial margin of the premaxillary

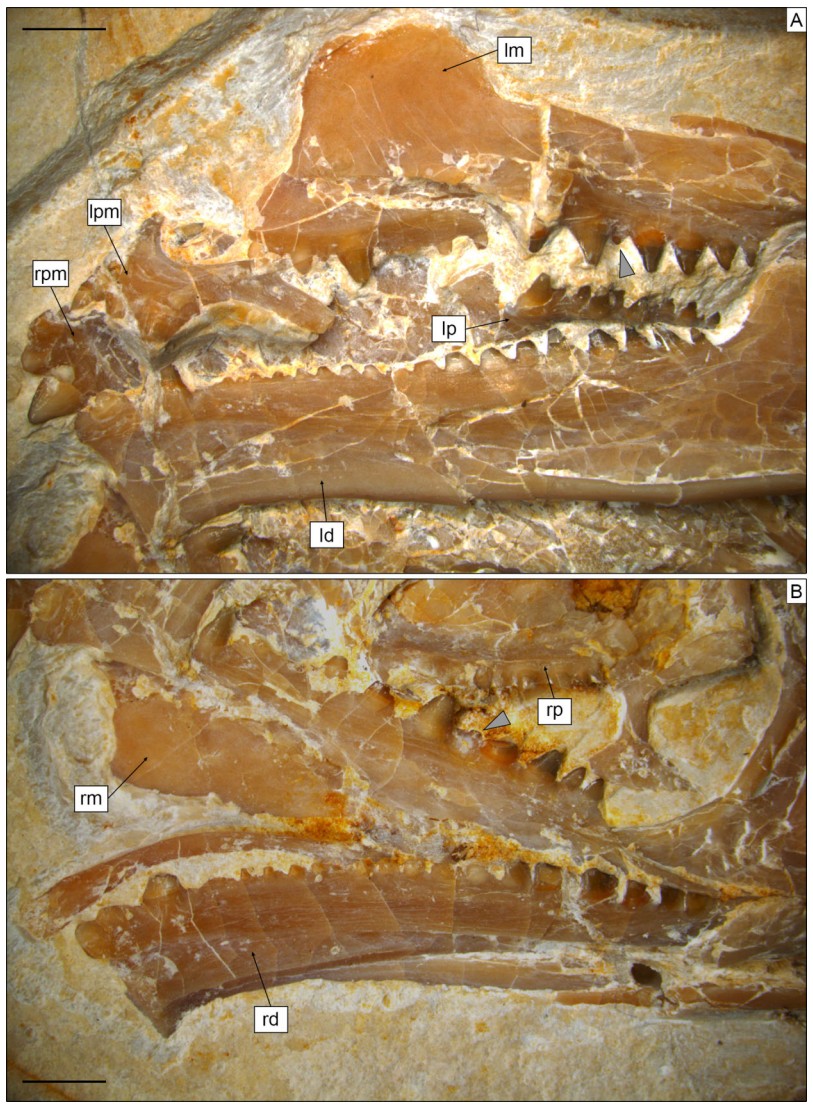

**Figure 4 Toothed elements of *Sphenofontis velserae* gen. et sp. nov.** (A) Left side of the skull, with the left maxilla (lm), the left palatine (lp), the left dentary (ld), and both left (lpm) and right (rpm) pre-maxillae. (B) Right side of the skull, with the right maxilla (rm), the right palatine (rp), and the right dentary (rd). Grey arrows point at the medially-displaced fourth additional maxillary teeth. Scale bars = 2 mm.                                 

body and the nasal process bears the smooth articulation surface with the opposed premaxilla. The anterior margin of the premaxilla is set at an angle of c. 70° towards the alveolar margin and curves very slightly posterodorsally. Dorsally, a narrow ascending nasal process projects from the premaxillary body. The distal part of the process is not visible, but it is clear from the left premaxilla that it narrows distally. The premaxilla also has a maxillary process that projects from the premaxillary body posterolaterally. This process set at a wide angle towards the alveolar border and tapers posterodorsally. In its posterodorsal portion, a wide, plate-like posteromedial process is present that would have been overlapped laterally by the maxilla in the articulated skull, as in *Clevosaurus*

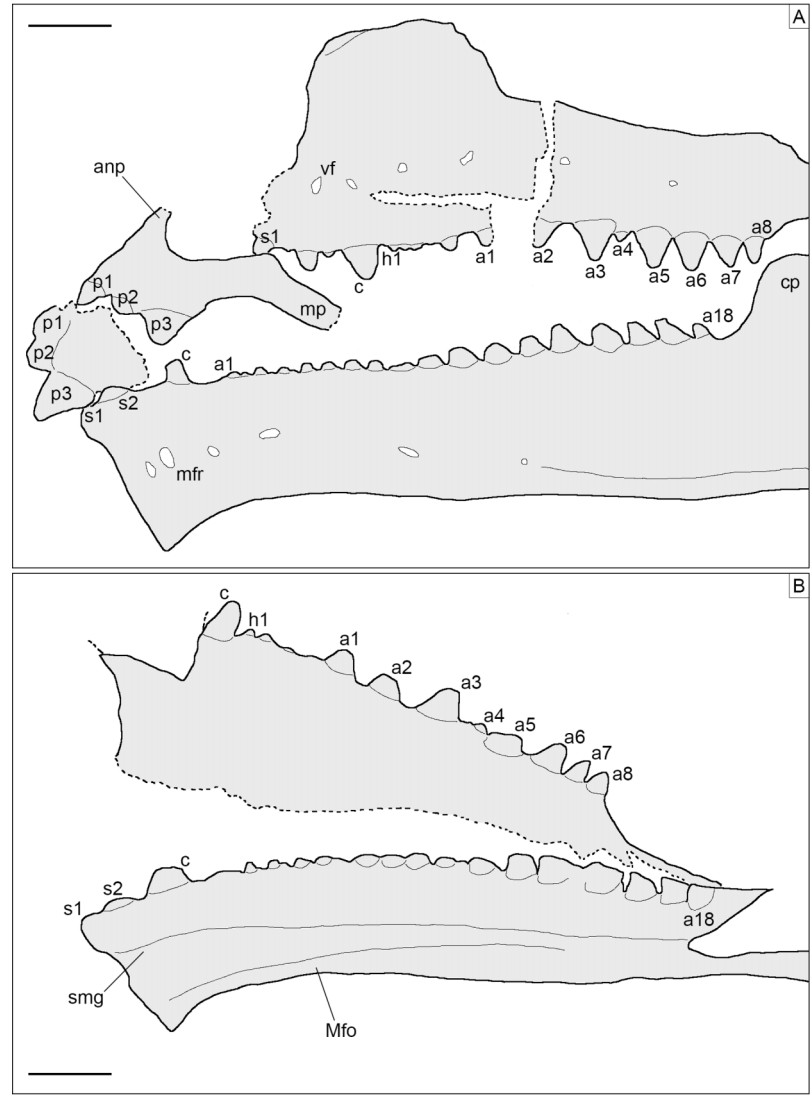

**Figure 5** **Toothed elements of *Sphenofontis velserae* gen. et sp. nov.** (A) Interpretative drawing of the marginal toothed elements on the left side of the skull, including the two premaxillae, the left maxilla, and the left dentary. (B) Interpretative drawing of the marginal toothed elements on the right side of the skull, including the right maxilla and the right dentary. Abbreviations: a1-8 + 18, additional teeth 1 to 8 and 18; anp, ascending nasal process; c, caniniform tooth; cp, coronoid process of the dentary; h1, hatchling tooth 1; Mfo, Meckelian fossa; mfr, mental foramen; mp, maxillary process; p1-3, premaxillary teeth 1 to 3, s1-2, successional teeth 1 and 2; smg, seconday medial groove; vf, ventrolateral foramen. Scale bars = 2 mm.               

(*Fraser, 1988*). However, in contrast to the latter taxon, this process is directed straight posteriorly and not posteroventrally, and it is not as robust as in clevosaurs. Together, the ascending nasal process and the maxillary process define the anteroventral margin of a moderately wide and anteriorly-located external naris. Although the maxillary process is long (longer than the premaxillary body), its distal end is not preserved, so it cannot be said with certainty whether the maxilla participated in the margin of external nares, as in *Sphenodon* (*Jones et al., 2011*), or if it was excluded from this margin by a premaxilla-nasal

contact posterior to that opening, as in *Clevosaurus* (*Fraser, 1988*; *Sues, Shubin & Olsen, 1994*), *Vadasaurus* (*Bever & Norell, 2017*) and *Priosphenodon* (*Apesteguía & Novas, 2003*).

The maxillae (Figs. 3, 4, 5) are elongated bones (but not as elongated as in *Pleurosaurus*), with a generally slender appearance, more than twice as long as high. The morphology of the anterior premaxillary process cannot be described as it is incompletely preserved in the left element (though it is possible that not much is missing) and not exposed in the right one. Nevertheless, it was clearly distinctly developed, in contrast with a small or absent process in Clevosauridae (*Sues, Shubin & Olsen, 1994*; *Bonaparte & Sues, 2006*; *Jones, 2006*) and an almost absent process in *Priosphenodon* (*Apesteguía & Carballido, 2014*) and *Sphenotitan* (*Martínez et al., 2013*). Just dorsal to the incomplete premaxillary process, the maxilla displays a slightly concave surface, which might have formed part of the external nares. The facial process is moderately low and wide; based on the left maxilla (which is almost completely preserved and more exposed than the right one), it extends for about 36% of the total length of the bone (5 mm out of about 14 mm). It is distinctly wider anteroposteriorly in *Priosphenodon avelasi* (*Apesteguía & Novas, 2003*) and considerably narrower in *Sphenodon* (A.Villa, 2019, personal observation; see also figures in *Evans, 2008*, and *Jones et al., 2011*), *Sigmala sigmala*, and *Pelecymala robustus* (see figures in *Fraser, 1986*). The process is dorsally convex, with subvertical anterior and posterior (orbital) margins and a slightly posterodorsally-sloping dorsal margin (Figs. 4A, 5A). Anterodorsally, the lateral surface of the process flexes distinctly medially, with a small vertical flange being present medially at its anterodorsal end. A small, posterodorsally- facing concavity above the short orbital margin most probably marks the contact with the prefrontal. The height of the process is roughly half that of the posterior (suborbital) process of the maxilla. *Cynosphenodon*, *Sphenodon*, and *Clevosaurus bairdi* have a distinctly higher facial process (*Sues, Shubin & Olsen, 1994*; *Reynoso, 1996*; *Jones et al., 2011*), whereas this process is almost absent in *Sphenotitan* (*Martínez et al., 2013*). The lateral surface is smooth. The posterior process is long, composing more than half of the length of the maxilla, and moderately robust. In lateral view, it is straight, with subparallel dorsal and ventral margins and a pointed posterior end. The orbital margin is straight to very slightly convex in its anterior half and slightly concave in the posterior portion. The posterior tip is bent laterally and overlaps the anteroventral part of the jugal, resulting in the formation of a short, but notable lateral shelf above the posterior end of the tooth row. A strongly developed medial process like the one displayed by maxillae of *Oenosaurus* (*Rauhut et al., 2012*) is absent. The lateral surface of the maxilla bears a row of ventrolateral foramina; the count of the latter is complicated by the preservation, but at least six of them seem to be visible on the left maxilla (being thus significantly more than in *Priosphenodon minimus* and *Sapheosaurus*; *Cocude-Michel, 1963*; *Apesteguía & Carballido, 2014*). Ventral to the row of foramina, there is a very shallow and narrow longitudinal groove. Anteriorly, below the facial process, this groove deepens, but broken walls indicate that this is due to breakage of an underlying channel within the bone, which opens in a large, anterolaterally facing foramen just 1 mm posterior to the anterior margin, at the level of the dorsal rim of the incomplete premaxillary process. Teeth are present along the

ventral margin, except for the posterior end of the posterior process and maybe also the anterior half of the premaxillary process.

The jugal (Fig. 3) is a very long and large bone, with a triradiate shape. The anterior and quadratojugal processes are slender, whereas the posterodorsal process is wider. The anterior process is long and tapers anterodorsally, forming part of the ventral border of the orbit. However, in contrast to *Clevosaurus* (*Sues, Shubin & Olsen, 1994*), *Priosphenodon* (*Apesteguía & Novas, 2003*), and *Oenosaurus* (*Rauhut et al., 2012*), the process does not extend to almost the anterior end of the orbit, but ends at about its mid-length, as in *Sphenodon* (*Jones et al., 2011*). The quadratojugal process is missing its distal tip on both sides of the skull, but on the right side the missing part probably did not extend much further, indicating that this process was distinctly shorter than the anterior one. Whether it contacted the quadratojugal and formed a complete jugal bar, as in many sphenodontians, cannot be said due to the incomplete preservation on both sides, but it seems likely, based on the relatively massive cross-section of the bone at its posterior break. Nevertheless, the presence of the quadratojugal process distinguishes SNSB-BSPG 1993 XVIII 4 from *Vadasaurus* (*Bever & Norell, 2017*). The dorsal portion of the posterodorsal process of the left jugal is hidden in the matrix, whereas the tip of the process of the right element is covered by the pterygoid wing of the quadrate, thus preventing evaluation of its complete length. The posterodorsal process is anteroposteriorly wide, plate-like and slightly posteriorly inclined. Thus, the ventral orbital margin curves into the posterior orbital margin in a wide angle, whereas the anteroventral margin of the infratemporal fenestra forms a sharp angle of approximately 70°. Both anterior and posterior processes of the jugal have a similar dorsoventral depth and are straight. The ventral margin of the jugal is thus straight. The smooth medial surface of the jugal is exposed on the right side. A small, anteroposteriorly elongate concave facet just below the orbital margin at the point where the ventral orbital margin curves onto the posterodorsal process probably represents the jugal articular facet for the ectopterygoid. The lateral surface is visible in the left element: it appears irregular, but this likely results from poor preservation and the surface was probably smooth as well originally (as indicated by some areas that appear less affected by the preservational status).

On the right side of the skull, an elongated, slightly curved rod of bone covering the anterior part of the posterodorsal process of the jugal represents the anterolateral process of the postorbital (Fig. 3), the tip of which almost reaches the ventral margin of the orbit. A clear expansion is visible at the dorsal base of this process, suggesting that the rest of the postorbital is still preserved, but largely covered by the pterygoid wing of the disarticulated right quadrate. However, the posterior margin of the orbit can be seen to continue dorsally, curving anteriorly in the last portion exposed, before this margin is covered by the collapsed elements of the palate, mainly the right pterygoid. Here, the dorsomedial end of the postorbital is visible as a bluntly rounded process that slots into a notch in the lateral margin of the postfrontal, as in *Sphenodon* (*Jones et al., 2011*), but unlike the situation in *Clevosaurus* (*Sues, Shubin & Olsen, 1994*) or *Vadasaurus* (*Bever & Norell, 2017*), in which the postfrontal flanks the dorsomedial process anteriorly. However, in contrast to *Sphenodon*, where the notch in the postfrontal is only visible in dorsal

view and a ventral sheet of bone covers the tip of the dorsomedial process of the postorbital ventrally (*Jones et al., 2011*), the peg-in-socket articulation between these two bones is here visible in ventral view. The dorsomedial process of the postorbital was shorter but slightly broader than the ventral process.

The postfrontal is largely covered by the pterygoid wing of the right quadrate and various palatal bones, so not much can be said about its detailed morphology. It was obviously a triradiate bone with a long anterior process that can be seen to flank the frontal laterally and thus forms part of the posterodorsal margin of the orbit and an equally long, pointed posterior process that flanked the anterior end of the parietal laterally, as in *Sphenodon* (*Jones et al., 2011*).

The rather well-preserved right quadrate is visible and mainly exposed in medial view. Of the left element, only the broad dorsal cotyle is exposed, while the rest of the bone is covered by the left mandible. The quadrate (Fig. 3) is dorsoventrally elongated. The pillar is straight and slender, occupying a very small portion of the width of the bone and expanding at both ends. It is slightly inclined posterodorsally in respect to the ventral condyles, indicating that the latter projected slightly posteroventrally in the articulated skull, as in *Sphenodon*, but unlike the rather straight and vertical quadrate in *Clevosaurus* (*Fraser, 1988*; *Sues, Shubin & Olsen, 1994*; *Sues & Reisz, 1995*) and *Vadasaurus* (*Bever & Norell, 2017*). The cephalic condyle is poorly preserved, but it is strongly widened anteroposteriorly and, based on the left element, also somewhat transversely. The mandibular articulation is also wide, expanding more mediolaterally than anteroposteriorly. Ventrally, it is split into two expanded condyles by a deep, V-shaped middle notch. The medial condyle expands slightly more ventrally than the lateral one. Both condyles are well rounded anteroposteriorly, the medial condyle more strongly than the lateral one. The posterior surface is deeply invaginated lateral to the quadrate pillar, with a small lateral flange extending from the latter laterally at the deep parts of this invagination. Lateral to this flange, a large quadrate foramen seems to have been present between the quadrate and quadratojugal, as in *Sphenodon* (*Jones et al., 2011*). Anteriorly, the pterygoid wing of the quadrate is developed as a long and wide bony lamina, which is offset from the ventral condyles by c. 1/4th of the height of the bone, but extends dorsally to almost the level of the cephalic condyle. It is tongue-shaped and almost as long (6.8 mm) as the quadrate is high (7.6 mm) and offset from the quadrate pillar and the ventral condyle by a notable step in medial view, resulting in a transversely broadened ventral margin of the wing in its proximal part.

The poorly preserved right quadratojugal (Fig. 3) is partially visible lateral to the related quadrate, contacting the latter both dorsally and ventrally. Quadrate and quadratojugal were almost certainly not fused dorsally, but the preservation does not allow an evaluation of a possible ventral fusion at the mandibular condyle. Nothing can be said about the lateral morphology or anterior extent of the quadratojugal, as these are hidden in the matrix below the quadrate.

Fragments of the squamosal (Fig. 3) are also visible in this area of the skull, dorsal and medial to the quadratojugal; a small portion of the squamosal is also visible on the left side of the skull. The small preserved portions include the parietal-squamosal contact on

the right side of the skull, in which a long, tapering medial process of the squamosal overlaps the parietal posteriorly and reaches almost the level of the basioccipital. The preserved section on this and the left side show that the medial squamosal bar was relatively slender, rod-like and posteriorly convex, as in *Sphenodon*.

The vomers are either not visible or not preserved. The right palatine (Fig. 3, 4B) is exposed and sufficiently preserved to be described in some detail, even though it is not complete. The exposed tooth row of the left palatine (Fig. 3, 4A) adds some additional information. The bone has an anteroposteriorly and transversely wide and laminar pterygoid process, which composes its main body. The posterior end of this process is broken off and the anterior end is not preserved. The bony lamina formed by this process is longer anteroposteriorly than wide transversely and seems to narrow somewhat anteriorly. The lateral margin of the preserved portion of the palatine bears a robust and very tall ridge, which carries a single row of palatine teeth (contra the presence of at least an extra median tooth in Clevosauridae, a cluster of median teeth in *Sphenotitan*, two rows in *Rebbanasaurus*, three rows in *Gephyrosaurus*, either two or three rows in *Planocephalosaurus*, and four rows in *Diphydontosaurus*; *Evans, 1980*; *Fraser, 1982*, *1988*; *Whiteside, 1986*; *Evans, Prasad & Manhas, 2001*; *Martínez et al., 2013*; *Hsiou, De França & Ferigolo, 2015*; *O'Brien, Whiteside & Marshall, 2018*; *Romo-de-Vivar-Martínez et al., 2021*). The tooth-bearing ridge of the left palatine is also exposed, being the only clearly visible portion of this element. The palatine tooth ridge is roughly parallel to the maxillary and dentary tooth-rows. The presence of an elevated palatine tooth ridge is in contrast with the palatine teeth of *Clevosaurus minor*, which are not elevated in a ridge (*Fraser, 1988*). The posterior end of the ridge seems to be continuous with a posterolateral suture with the ectopterygoid. There is no indication of an opening between the palatine and ectopterygoid, as it is present in *Sphenodon* (*Jones et al., 2011*) and *Oenosaurus* (*Rauhut et al., 2012*). However, it should be noted that it cannot be completely ruled out that the palatine has been slightly shifted and compressed onto the ectopterygoid. The anterior end of the left palatine shows that a narrow shelf was present lateral to the toothed ridge, with a short, tapering anterior process for the contact with the maxilla, as in *Sphenodon* (*Jones et al., 2011*).

The pterygoids (Fig. 3) are large and long bones, with an overall slender appearance: that is, the length of all branches is strongly higher than their maximal width, as far as can be seen from the preserved portions. Both pterygoids are incompletely preserved, but the right one is in a better condition and more exposed. The palatine process is fragmentary and not completely visible in both elements. Nevertheless, it appears very long, with a rather narrow base and expanding slightly at about its midlength. The lateral margin of this process contacts the right palatine for the entire length of the preserved portion of the latter, whereas the medial margin comes in contact with the opposed pterygoid just anterior to a moderately small, deltoid interpterygoid vacuity that is only slightly longer than its maximal width. As far as can be judged from the poor preservation, the ventral surface of the palatine process is smooth, without teeth (in contrast to *Brachyrhinodon, Diphydontosaurus, Gephyrosaurus, Planocephalosaurus, Polysphenodon, Sphenotitan,* and *Clevosaurus*; *Evans, 1980*; *Fraser, 1982*, *1988*; *Whiteside, 1986*; *Fraser &*

*Benton, 1989*; *Bonaparte & Sues, 2006*; *Jones, 2006*; *Martínez et al., 2013*; *Hsiou, De França & Ferigolo, 2015*; *O'Brien, Whiteside & Marshall, 2018*). The pterygoid flange is short, straight to very slightly flexed posteriorly and laterally directed. The quadrate process is long and rod-like in ventral view, and straight. It narrows distally.

The posteromedially-directed basipterygoid fossa is visible by the base of the latter process. The fossa received the basipterygoid process of the sphenoid, which was clasped anteromedially by a short and more tubercle-like (compared to the other pterygoid branches) process of the pterygoid. Roughly in the same area, at the meeting point of the three branches composing the pterygoid, a ventral bony expansion is visible, which is short and ventrally rounded.

The right ectopterygoid (Fig. 3) is well preserved and exposed. It seems to be still in articulation with at least the pterygoid (and maybe the palatine), but displaced from the maxilla. It is a small and very slenderly-built bone, with a complex shape. It has a straight and narrow middle portion, expanding at both ends. The medial end displays a long, narrow, but bulbous and ventrally raised posteroventral projection that contacts the distal end of the pterygoid flange of the pterygoid. Dorsal to this, the ectopterygoid has another, anteromedial expansion that likely covered the flange on the dorsal side. The lateral end of the ectopterygoid has a triangular shape in ventral view (unlike the laterally-forked ectopterygoid of *Oenosaurus*; *Rauhut et al., 2012*), with a posterior projection that is slightly longer than the anterior one. The ventral surface of the lateral side of the ectopterygoid is smooth, with no ventral projections, and its lateral margin is straight or slightly convex.

The different bones composing the braincase are unfused. This holds true for all elements that are at least partially visible (i.e., basioccipital, sphenoid, prootic, exoccipital, and opisthotic), but cannot be evaluated for the supraoccipital, which is not exposed due to the specimen resting on its dorsal side; however, the slight disarticulation of the braincase elements indicates that this element was also unfused. The most clearly visible elements of the braincase are the sphenoid and the basioccipital. Other elements are preserved as well, but are only partially exposed and less well-preserved.

The basioccipital (Fig. 3) is small and subpentagonal in outline in ventral view. It is slightly wider than long and widens gradually from the base of the occipital condyle towards the contact with the sphenoid. The ventral surface is flat and smooth between the well-developed basal tubera, which are located at the anterolateral sides of the basioccipital. The basal tubera are widely separated, narrow and project well ventrally, similar to the condition in *Sphenodon* (*Evans, 2008*), but unlike the broader and less conspicuous tubera in *Oenosaurus* (*Rauhut et al., 2012*). They are mainly composed by the basioccipital, with only a small anterior contribution by the sphenoid. As in *Sphenodon*, the anterior end of the basioccipital slots into a wide concavity on the posterior side of the sphenoid, but the anterior expansion of the basioccipital is smaller than in this taxon and anteriorly rounded rather than angular (see *Evans, 2008*). Posteriorly, the occipital condyle is almost completely composed by the basioccipital. The condyle is approximately as wide as the space between the basal tubera and has a straight (i.e., not notched) posterior margin. It is separated from the main body of the basioccipital by a marked step, but a constricted neck

is absent. In lateral view, the condyle is level with the floor of the basioccipital and sphenoid.

The sphenoid (Fig. 3) is longer than the basioccipital. It has a flat and smooth ventral surface, similar to the *Homoeosaurus maximiliani* specimen stored in the Teyler Museum in Haarlem (specimen n. 3955 in *Cocude-Michel, 1967b*) and unlike the concave surface seen in *Oenosaurus* (*Rauhut et al., 2012*) and *Sphenodon*. The posterior margin of this bone is strongly concave for the contact with the basioccipital, and the posterolateral corners of the sphenoid are slightly raised for the contact with the basal tubera on the basioccipital. From these processes, the ventral side of the sphenoidal body constricts gradually towards the base of the basipterygoid processes. Anteriorly, the sphenoid bears a rather long parasphenoid rostrum, the complete length of which cannot be evaluated. However, it extended considerably further anteriorly than the basipterygoid processes. The rostrum is located between two moderately short and thick basipterygoid processes, unlike the longer and narrower processes of *Clevosaurus brasiliensis* (*Hsiou, De França & Ferigolo, 2015*), although they seem to be slightly longer and more anteriorly directed than in *Sphenodon* (*Evans, 2008*). The processes expand slightly at their distal ends, which contact the respective pterygoid in the basipterygoid fossa. On the ventral surface of the sphenoid, two wide and elliptical foramina are present by the base of the basipterygoid processes, in the same position as the Vidian grooves in *Sphenodon* (*Evans, 2008*); these foramina thus most probably represent the ventral entrances of ossified Vidian canals. Some other small and more circular foramina are also present posterior to the two elliptical ones and along the midline of the bone, some of them being located in a shallow fossa placed in the middle of the ventral surface of the bone. The lateral margins of the sphenoid expand anterodorsally towards well-developed supravenous processes and posterolaterodorsally to give rise to long, narrow and laterally-pointed alar processes contacting the prootics, similar to the condition in *Clevosaurus* (*Fraser, 1988*). The latter bones are too poorly preserved to reveal much useful morphological information. The disarticulated right prootic shows the incisura prootica (exit of the trigeminal nerve), which is developed as an anterodorsally opening incision in its anterior margin, similar to the condition in *Clevosaurus* (*Fraser, 1988*) and *Sphenodon*, although the incisura seems to be relatively smaller than in the latter taxon (*Evans, 2008*).

The preservation is a little bit better for the exoccipital and opisthotic (Fig. 3), at least on the left side of the cranium. These bones are unfused in SNSB-BSPG 1993 XVIII 4, which therefore lacks a fused otooccipital. The left exoccipital is well-preserved, but disarticulated from the basioccipital into the horizontal plane by compression. The exoccipitals are roughly triangular in outline, with a wide ventral base. The posteroventral edge of the bone is slightly expanded posteriorly and rounded and formed a small portion of the dorsolateral part of the occipital condyle. The medial margin, which formed the lateral edge of the foramen magnum, is only slightly concave. The lateral margin runs dorsolaterally upward at a roughly 45° angle. The dorsal margin of the exoccipital is quite narrow anteroposteriorly, but expanded transversely, forming a transversely very slightly convex articular facet for the supraoccipital. Three hypoglossal foramina seem to be present. They are placed in the ventrally expanding lateroventral side of the exoccipital, with the medialmost

foramen being the most anteriorly placed and smallest and the other two foramina being consecutively larger and placed more posterolaterally. The opisthotic is less well-preserved and the only feature that can be confidently described is a moderately short but well-developed paroccipital process. It was not possible to locate the stapes, which may be lost.

The lower jaws are rather well preserved. They are not as deep as in eilenodontines (*Rasmussen & Callison, 1981*; *Apesteguía & Novas, 2003*; *Martínez et al., 2013*; *Apesteguía & Carballido, 2014*), but rather low and elongate, with a marked coronoid process, as in the vast majority of rhynchocephalians. The left mandible is exposed in lateral view, whereas the right one shows its dorsomedial side. The portion posterior to the tooth row is not as short as in *Sphenovipera* (*Reynoso, 2005*), but more comparable to most rhynchocephalians, such as *Sphenodon*. The dentary (Figs. 3, 4, 5) is very long, making up about 83% of the lower jaw (25 mm out of 30 mm). These proportions recall those found in all other rhynchocephalians. It is slightly less slender than that of *Cynosphenodon* (*Reynoso, 1996*), *Sphenocondor* (*Apesteguía, Gómez & Rougier, 2012*), cf. *Diphydontosaurus* sp. from Vellberg (*Jones et al., 2013*), *Tingitana*, and the "sphenodontian B" from the Moroccan site of Anoual (*Evans & Sigogneau-Russell, 1997*). In lateral view, it is rather straight, with a sinusoidal ventral margin, being slightly concave in its anterior third and slightly convex over the posterior two thirds (unlike the generally convex margin in *Priosphenodon* and *Kawasphenodon expectatus*; *Apesteguía & Novas, 2003*; *Apesteguía, 2005*; *Apesteguía & Carballido, 2014*). The anterior end is very slightly deflected ventrally and bends slightly medially. It bears a high mandibular symphysis, with an upside-down teardrop-shaped surface. The symphysis is steeply inclined at approximately 70° towards the horizontal, unlike the more obliquely oriented symphysis in *Oenosaurus* (*Rauhut et al., 2012*), *Pamizinsaurus* (*Reynoso, 1997*), or *Cynosphenodon* (*Reynoso, 1996*). Anteroventrally, a small ventral expansion creates a small "chin", as seen in many rhynchocephalians. Due to the more vertical orientation of the symphysis, the projection is not as posteriorly located as in *Pamizinsaurus* (*Reynoso, 1997*). On the medial side, the dentary has a narrow Meckelian fossa, which is very shallow in the anterior half of the bone but deepens posteriorly. The fossa is positioned on the ventral side of the anterior part of the dentary, but is not closed by the expansion of the ventral margin as it is in *Gephyrosaurus* (*Evans, 1980*). A second groove (secondary medial groove sensu *Reynoso, 1996*) is also present in the anterior part of the dentary, dorsal to the shallow portion of the Meckelian fossa. This second groove starts from the Meckelian fossa at about the level of the half-length of the dentigerous portion of the dentary posteriorly and runs anterodorsally. It is very shallow, becoming even more shallow (almost indistinguishable) towards the anterior end of the dentary. It reaches the symphysis, being recognizable in lateral view as a very shallow notch between the symphyseal facet and the first dentary tooth and as a notable incision in the medial margin of the dorsal part of the symphysis in medial view. A similar notch is present both in extant *Sphenodon* and some fossil rhynchocephalians as well (*Evans, Prasad & Manhas, 2001*; *Jones et al., 2009b*). The secondary medial groove was considered diagnostic for *Cynosphenodon huizachalensis* by *Reynoso (1996)*, but we can confirm its presence at least in both the Brunn specimen and the extant *Sphenodon* (A.Villa, 2019, personal observation). The lateral surface of the dentary displays a moderately wide longitudinal groove,

marked dorsally by the development of secondary bone (a feature related to derived rhynchocephalians; *Apesteguía, Gómez & Rougier, 2012*). This lateral groove appears distinctly shallow in most of the bone, even though the crushing of the specimen gives it a deeper appearance in the posterior portion; it seems to disappear below the coronoid process. The groove hosts some mental foramina. A confident count of the latter is difficult, but at least six of them seem to be visible. There is no striation on the ventrolateral surface of the dentary, in contrast with *Pleurosaurus* and opisthodontians (*Cocude-Michel, 1963*, *1967a*; *Apesteguía, Gómez & Rougier, 2014*; A.Villa, 2019, personal observation) and probably also *Clevosaurus brasiliensis* (*Hsiou, De França & Ferigolo, 2015*; fig. 4A). The dorsal margin of the dentary bears the teeth (Figs. 4, 5). The latter are not limited to the posterior end of the tooth row, as in *Kawasphenodon* (*Apesteguía, 2005*). The tooth bearing portion of the dentary is significantly shorter in *C. brasiliensis*, when compared to *Sphenofontis* (*Hsiou, De França & Ferigolo, 2015*). Towards its posterior end, the dentary of SNSB-BSPG 1993 XVIII 4 develops a dorsally-directed coronoid process, which is anteroposteriorly wide and lower than the depth of the dentary anterior to the process (in contrast to *Oenosaurus*; *Rauhut et al., 2012*), and a posteriorly-directed inferior posterior process, which is dorsoventrally deep and long. The coronoid process is dorsally straight to slightly concave and generally similar to the coronoid process in *Sphenocondor* (*Apesteguía, Gómez & Rougier, 2012*), with its posterior third being formed by the surangular. The inferior posterior process seems to end in a posteriorly-pointed tip between the surangular and the angular, although the distal end of the laterally-exposed left dentary is covered by the jugal. A large, anteroposteriorly-elongated mandibular foramen is developed as a marked posterior incision between the two processes in lateral view. The presence of an enlarged mandibular foramen is considered to be a synapomorphy of sphenodontians (*Rauhut et al., 2012*), but it appears not to be present either in *Tingitana anoualae* or in the Moroccan "sphenodontian B" (*Evans & Sigogneau-Russell, 1997*). In SNSB-BSPG 1993 XVIII 4, the posterior process of the dentary is longer than the base of the coronoid process, whereas this process is as long as the base of the coronoid process in *Sphenocondor* (*Apesteguía, Gómez & Rougier, 2012*). Its posterior end reaches the level of the posterior half of the mandibular articulation, as in *Sphenodon* and other derived rhynchocephalians (*Evans, 2008*; *Rauhut et al., 2012*).

There is no splenial. The coronoid, which is visible only on the right side (Fig. 3), is an anteroposteriorly-elongated bone on the medial side of the coronoid process, straight in dorsal view. The coronoid has a very short anteromedial process, which fits in a distinct articular surface on the medial surface of the dentary, and a longer posterior process. A low and rather wide (dorsal) coronoid process is also present; it is dorsally narrowly rounded. In the left mandible, this rounded tip protrudes dorsally on the medial side of the dentary coronoid process, similar to the condition in *Cynosphenodon* and *Sphenodon*, in which, however, the dorsal tip of the coronoid is more pointed (*Reynoso, 1996*; *Evans, 2008*). The surface of this dorsal process of the coronoid differs from most other bone surfaces and seems to be more calcitic, which usually indicates preservation of cartilagenous structures or connective tissue in the southern German plattenkalks (*Tischlinger & Unwin, 2004*). The coronoid is considerably higher in *Oenosaurus* than in SNSB-BSPG 1993 XVIII 4 (*Rauhut et al., 2012*). A discrete coronoid was reported as

lacking in *Clevosaurus hudsoni* (*Fraser, 1988*; *O'Brien, Whiteside & Marshall, 2018*), but it was recently described in fossils referred to this species by *Chambi-Trowell, Whiteside & Benton (2019)*. The angular (Fig. 3) is elongated and strip-like. It has a pointed anterior end on the medial side of the dentary and an enlarged, rounded posterior end on its lateral side. The angular extends from about the level of the 14th dentary tooth, or two fifths of the length of the lower jaw, to approximately the level of the start of the retroarticular process. Articular, prearticular, and surangular appear to be fused in a single compound bone (Fig. 3), which is relatively short compared to the overall length of the lower jaw, accounting for c. 13 mm of the total length of 30 mm. Medially, a deep, anteroposteriorly-elongated and rather wide adductor fossa is present between the coronoid and the jaw articulation (unlike the reduced fossa in *Sphenovipera*; *Reynoso, 2005*). The articular condyle is wide and subquadrangular in dorsal view. It is crossed longitudinally by a robust and well-developed ridge, which fits in the notch of the mandibular condyle of the quadrate and splits this condyle into two portions. The medial portion is deeper and wider than the lateral one; whereas the latter is transversely straight, the former is slightly concave. Anterodorsally on the lateral surface, the surangular forms the posterior part of the coronoid process and defines the posterior margin of the mandibular foramen. The posterior end of the compound bone (and thus of the lower jaw as a whole) forms a thick retroarticular process, which has a subtriangular shape and a truncated posterior end. The retroarticular process is rather short, its anteroposterior length being similar to that of the articular condyle. The lateral margin of the process is flat to slightly convex, whereas the medial edge is concave. The dorsal surface of the retroarticular process houses a marked, transversely concave depression. The retroarticular process is longer and more slender in pleurosaurids (*Cocude-Michel, 1963*; *Bever & Norell, 2017*).

In addition to the various bones or bone fragments that likely represent part of the skull roof, the palate, and the braincase, there are two elongated bones of difficult interpretation. The first one is a rod-like bone that overlies the quadrate process of the left pterygoid, but is covered by the left dentary anteriorly and to some degree by the prootic posteriorly (anterior and posterior are referred only in relation to the position of the skull ends here and not to the actual ends of the so-far unrecognized bone). The rod is narrow, but expands distinctly close to the prootic. The shape of this bone is somewhat reminiscent of the epipterygoid, but two aspects speak against its interpretation as such: first, the fact that it appears too narrow in what should be its dorsal portion, without expansion towards its dorsal end; and second, the position ventral to the pterygoid. This position could be more consistent with an interpretation of this bone as part of the hyobranchial skeleton. At the moment, however, a confident identification is not possible. The other indeterminate bone is exposed between the anterior half of the right dentary and the right maxilla. It appears as an elongated, narrow and curved bone, but it is not clear how much of it is still hidden in the matrix. This bone is most probably the ceratohyal.

**Dentition.** Teeth (Figs. 4, 5) are present on the premaxillae, maxillae, palatines, and dentaries (in contrast to the edentulous *Piocormus* and *Sapheosaurus*; *Cocude-Michel,*

*1963*; *Fabre, 1981*). All teeth are acrodont (sensu *Evans, 2008*), as in most sphenodontians, but unlike the pleurodont teeth present in *Diphydontosaurus*, *Gephyrosaurus*, *Whitakersaurus*, and the Vellberg cf. *Diphydontosaurus* sp. (*Evans, 1980*; *Whiteside, 1986*; *Heckert et al., 2008*; *Jones et al., 2013*). All teeth are conical, being also somewhat mediolaterally compressed. Teeth are not pleuracrodont (sensu *Whiteside & Duffin, 2017*), as in *Deltadectes* (*Whiteside, Duffin & Furrer, 2017*). The dentition is markedly heterodont. Except for the premaxillary teeth and the successional teeth on the dentary, all teeth are well spaced.

Each premaxilla bears three teeth, which are slightly less compressed than those of other tooth-bearing bones. The most lateral tooth is distinctly larger than the other two and clearly isolated from them. The mesialmost tooth is the smallest tooth in the premaxilla. The two mesial teeth are coalesced at their base. The distal tooth displays a rounded tip and low and sharp carinae mesially and distally. Very low striae are (poorly) visible on the exposed lingual side of this tooth, being oriented vertically. The tips of the smaller teeth are eroded, but they display clear flanges at the sides. The most medial tooth has a flange only laterally, whereas the other tooth has flanges on both sides. These flanges are robust and not sharp; the one of the medialmost tooth fuses with the medial flange of the other tooth, resulting in the coalescent morphology of this part of the premaxillary dentition. A very poorly distinct vertical striation is visible on the lingual surface of this tooth as well.

The maxillary dentition of SNSB-BSPG 1993 XVIII 4 can be split into three different sections, as described for other derived rhynchocephalians. At the anterior end of the bone, several successional teeth are present (in contrast to *Sigmala* and *Pelecymala*, which lack maxillary successional teeth; *Fraser, 1986*). The exact number of these teeth cannot be confidently counted, due to the anterior end of both maxillae being (at least partially) covered by other bones. On the left side, at least four successional teeth are visible, but a fifth one was probably present between the first and second preserved ones. The posteriormost of these teeth is considerably larger (caniniform) than the preceding ones, as in *Cynosphenodon* (*Reynoso, 1996*) and *Sphenodon* (*Robinson, 1976*; *Evans, 2008*). Posterior to this section, there is a short row of very worn, small, and poorly preserved hatchling teeth. The total number cannot be securely counted in this case either, but four teeth can be estimated for both maxillae. Following the hatchling section is a long row of additional teeth, including eight teeth on both sides. These teeth show an increase in size posteriorly, reaching maximum size with the third tooth in this section. Distal to this, there is a very small fourth tooth and then a fifth tooth that is slightly smaller than the third, which again is followed by a decreasing trend in tooth size. The fourth tooth is similar in size or even smaller than the posteriormost maxillary tooth and appears medially displaced compared to the main axis of the tooth row. A trend similar to that involving tooth size is recognizable in tooth width, with the third tooth having the widest tooth base with successively more narrow teeth both anteriorly and posteriorly (again, with tooth four as an exception). None of the maxillary teeth bears either distinct flanges or a developed striation on the exposed labial surface, although a sharp, carina-like edge seems to be present on both the mesial and distal edges lingually, separating a rather flat lingual from a mesiodistally convex lateral side. The tooth tip appears blunt to rather rounded

in most of the preserved teeth, most probably due to wear. In total, at least 15 teeth can be counted on the maxilla.

At least eight (right) or nine (left) palatine teeth are present. These are conical and both smaller and narrower than the related maxillary teeth. They are distributed along a single axis and show a posteriorly-decreasing trend in size, with the largest tooth at the anterior end of the row. The tip is rounded. The general morphology of the palatine teeth is rather simple, with no flanges and no evident ridges of striation. In contrast, small flanges are present in *C. hudsoni*, *Opisthias*, *Priosphenodon*, *Sphenodon*, and *Godavarisaurus* (*Evans, Prasad & Manhas, 2001*; *Apesteguía & Carballido, 2014*; *Hsiou, De França & Ferigolo, 2015*), whereas *Planocephalosaurus*, *Rebbanasaurus*, and the indeterminate Brazilian sphenodontian MMACR-PV-051-T have striated teeth (*Fraser, 1982*; *Evans, Prasad & Manhas, 2001*; *Romo-de-Vivar-Martínez et al., 2021*). Proportionally, palatine teeth are not as large as in e.g., *Clevosaurus hudsoni* (*Fraser, 1988*).

As in the maxillae, the dentary dentition also includes few successional teeth, unlike *Sigmala* (*Fraser, 1986*). Three successional teeth are present in SNSB-BSPG 1993 XVIII 4, in contrast with one in *Opisthias* and five in e.g., *Rebbanasaurus* (*Gilmore, 1910*; *Evans, Prasad & Manhas, 2001*). The successionals of the Brunn specimen include two low and rounded teeth (likely due to wearing) at the anterior end of the dentary and a larger (caniniform) one posterior to the former. The third tooth displays a low carina at least on the mesial side; the possible presence of a similar carina on the distal side cannot be evaluated, however. The two anterior successional teeth are located very close to each other (almost coalescing), whereas the third is isolated from them by a notable gap. It is also separated from the teeth located posterior to it by an even larger space that probably indicates the original position of the hatchling dentition. *Cynosphenodon* also possesses an isolated and large caniniform tooth located roughly in the same place of the dentary tooth row, which is both preceded and followed by ridge-like portions of the row (*Reynoso, 1996*). *Sphenovipera* has (at least) two caniniforms, which further differ from the single one seen in SNSB-BSPG 1993 XVIII 4 because of the presence of dorsoventral grooves on the anterior surface (the supposed venom apparatus hypothesized by *Reynoso, 2005*). Two caniniform dentary teeth are present in *Theretairus* as well (*Simpson, 1926*). Distal to the successional series of SNSB-BSPG 1993 XVIII 4 is a long row of triangular teeth that increase distinctly in size posteriorly, starting from very small ones anteriorly. The large teeth in the posterior section are similar in size to those in the posterior section of the maxilla, but they don't reach the size of the largest maxillary tooth. The largest dentary teeth are either the fourth or the fifth starting from the posterior end of the row. As in the maxillae, tooth width follows a pattern that recalls that of the size. The widest/largest teeth on the dentary display moderately developed flanges mesially and distally, with the mesial one being better developed. Less developed flanges are present in smaller teeth also, at least in the posterior portion of the row with larger teeth. The flanges have a mesiolingual to distolabial course. Striae are present on the lingual surface of the anteriormost tooth (first tooth of the successional series), but they are apparently absent in all of the other teeth. The labial surface is always unstriated. Total tooth count is 21 in the dentary of SNSB-BSPG 1993 XVIII 4.

 

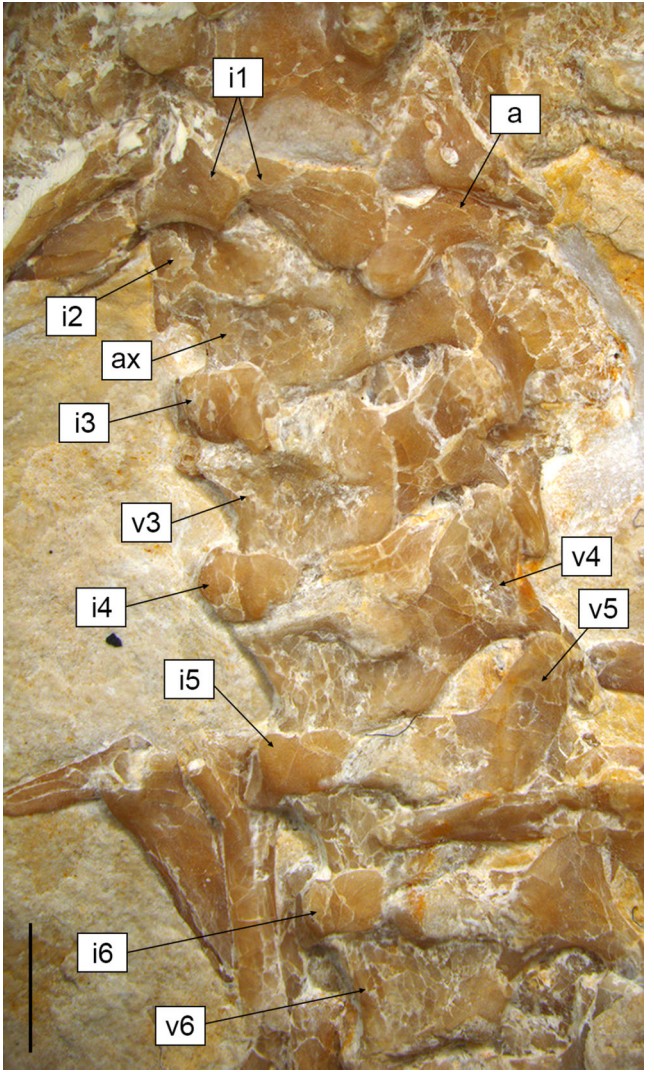

**Figure 6 Cervical region of *Sphenofontis velserae* gen. et sp. nov.** Scale bar = 2 mm. Abbreviations: a, atlas; ax, axis; i1-6, first to sixth intercentra; v3-6, third to sixth vertebrae.

**Axial skeleton.** The total number of vertebrae that can be counted is 66. Of these, 25 are presacrals (Figs. 6, 7), two are sacrals (Fig. 8), and 39 are caudals (Figs. 8, 9). The presacral vertebral count recalls *Sphenodon* (*Hoffstetter & Gasc, 1969*; *Fabre, 1981*) and is higher than in *Homoeosaurus maximiliani*, *Kallimodon*, *Leptosaurus*, *Piocormus*, and *Sapheosaurus* (*Cocude-Michel, 1963*, *1967b*; *Fabre, 1981*). The posteriormost caudal vertebra is in posterior continuity with a long and thin strip of calcified tissue that likely represents a regenerated posterior end of the tail (Fig. 9). The regenerated portion makes up roughly 19% of the total tail length (approximately 43 mm out of 221 mm). The tail is roughly twice as long as the body anterior to the first caudal vertebra (see measurements in Table 1). It is longer than in *Homoeosaurus solnhofensis*, both in terms of number of caudal vertebrae and of relative length compared to the rest of the body (*Cocude-Michel, 1963*; *Fabre, 1981*). SNSB-BSPG 1993 XVIII 4 has distinctly many fewer

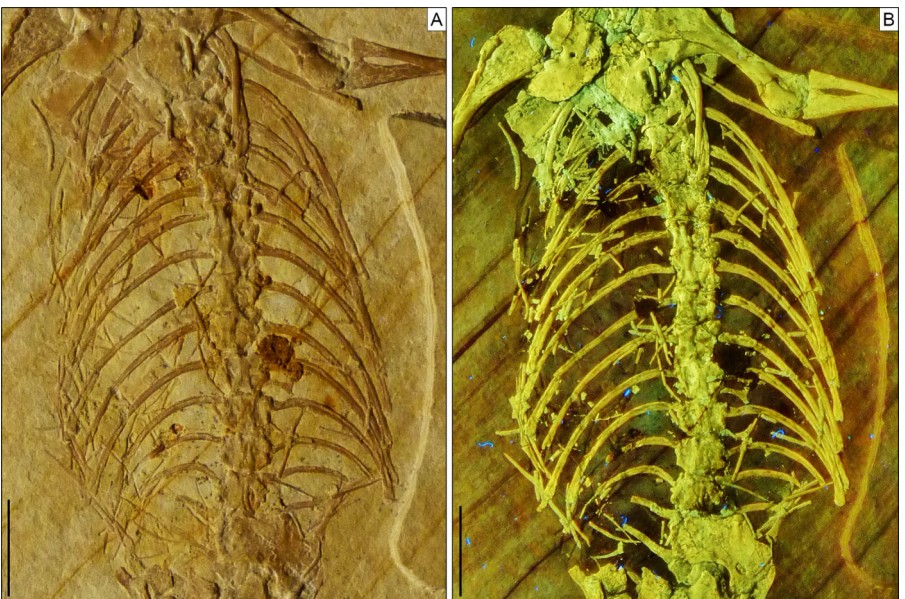

**Figure 7 Trunk region of *Sphenofontis velserae* gen. et sp. nov.** (A) Standard light; (B) UV-light. Scale bars = 1 cm. The UV-light photo in B was taken by Helmut Tischlinger.

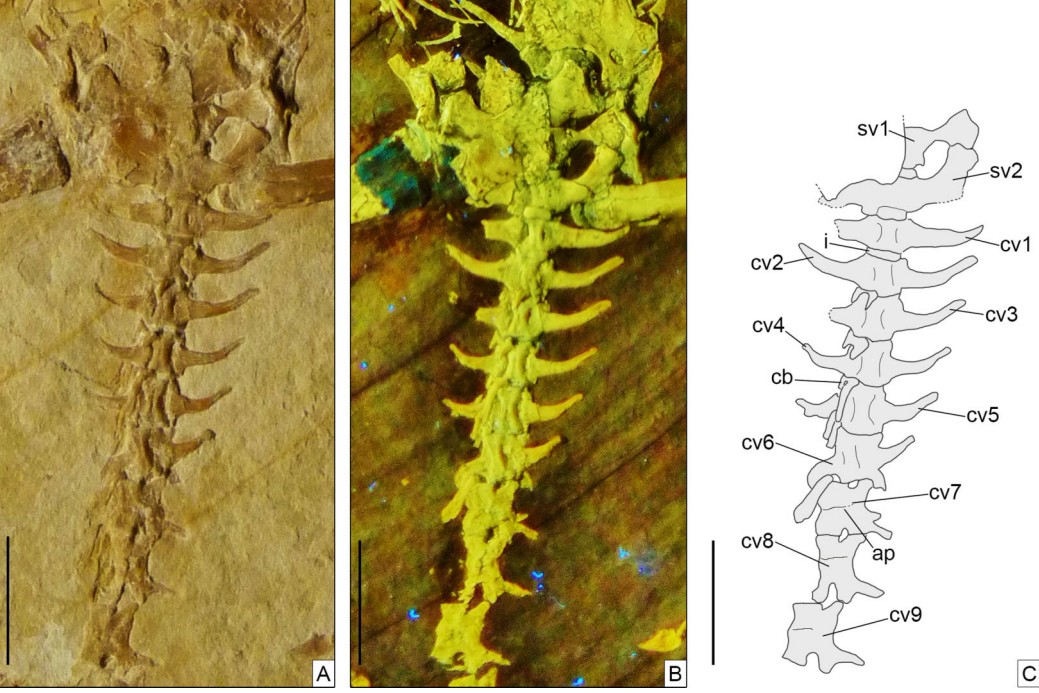

**Figure 8 Sacral and anterior caudal region of *Sphenofontis velserae* gen. et sp. nov.** (A) Standard light; (B) UV-light; (C) interpretative drawing. Scale bars = 1 cm. Abbreviations: cb, chevron bone; cv1-9, first to ninth caudal vertebrae; i, intercentrum; sv1-2, first and second sacral vertebrae. The UV-light photo in B was taken by Helmut Tischlinger.

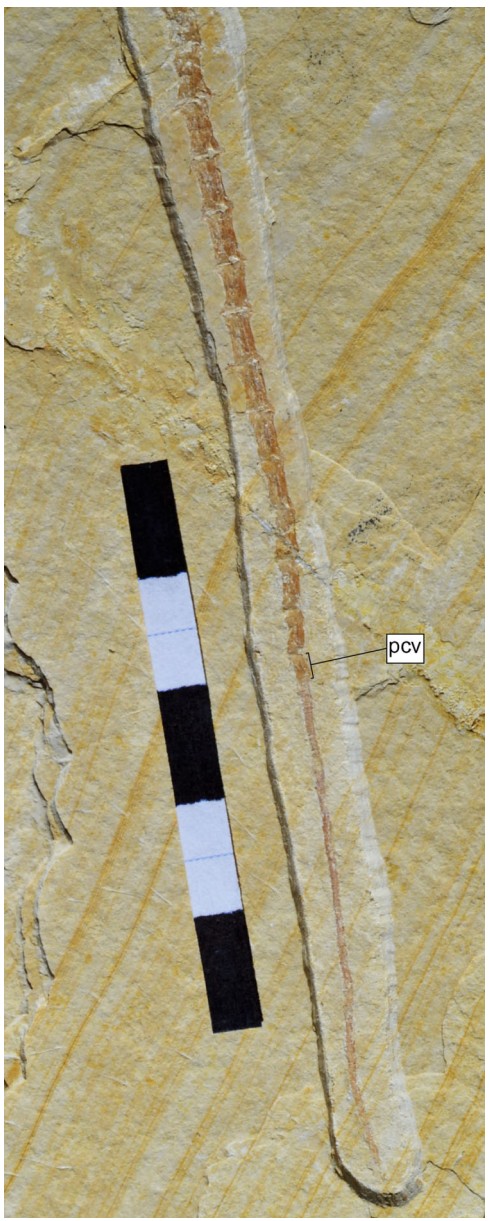

**Figure 9 Distal end of the tail of *Sphenofontis velserae* gen. et sp. nov.** The most posterior caudal vertebra (pcv) is shown, followed by the regenerated portion of the tail. Each subdivision of the scale bar is 1 cm.

vertebrae than the extremely elongated marine *Pleurosaurus* (*Cocude-Michel, 1963*, *1967a*; *Fabre, 1981*), whereas it has two more presacral vertebrae and, considering the regenerated portion, likely also more caudal vertebrae than *Vadasaurus* (*Bever & Norell, 2017*). The axial skeleton is not pachyostotic.

The proatlas, if present, is not visible in SNSB-BSPG 1993 XVIII 4. The first intercentrum is visible (Figs. 3, 6). It is broken into two portions. This intercentrum is narrower in the middle, but expands towards the sides. The element is ventrally convex. A narrow and elongated concave surface runs the entire posterior margin, being visible in

ventral view. The posterior margin itself is concave in ventral view. On the left side, part of the neural arch of the atlas is exposed (Figs. 3, 6), showing concave anterior and posterior margins and a short dorsal posterior process. The anterodorsal edge is overlain by the exoccipital, so it cannot be said if a pronounced anterior process was present, as is the case in *Sphenodon* (*Jones et al., 2009a*), *Gephyrosaurus* (*Evans, 1981*), and *Planocephalosaurus* (*Fraser & Walkden, 1984*). The axis and most of the subsequent exposed presacral vertebrae are visible in left ventrolateral view (Figs. 3, 6). The axis is rather short and slightly thinner than the following cervical vertebrae. The rather massive second intercentrum is recognizable, extending ventrally from the axis. A suture line is clearly visible between this intercentrum and the centrum of the axis, which are therefore unfused. The anterior end of the centrum expands ventrally to cover the intercentrum posteriorly. The axis centrum has a ventrally concave ventral margin. The neural arch is completely fused with the centrum and displays a small and circular fossa at its base, located in the middle of the lateral wall. No diapophyseal lateral protuberance seems to be present. The rather long left postzygapophysis is exposed, as is part of the neural spine. The latter is at least as high as the neural arch of the following cervical and projects posteriorly up to the midlength of the following vertebra.

Postaxial presacral vertebrae (Figs. 3, 6, 7) start with a size that is comparable with that of the axis, but then gradually enlarge posteriorly. The centrum length is roughly doubled in the posteriormost exposed presacrals when compared to the axis. The centra are hourglass-shaped, with concave ventral and lateral margins. There is no sign of a condyle, neither anteriorly nor posteriorly, thus suggesting amphicoelous vertebrae (even though this cannot be clearly confirmed due to articulation of the vertebrae). A ventral keel is present throughout the entire vertebral column, being sharper in anterior vertebrae and stouter posteriorly. The neural arch has lateral walls with concave anterior and posterior margins and long zygapophyses. The arch is either as high or slightly higher than the centrum. It becomes larger in more posterior vertebrae, following the general increase in size shown by the vertebrae. An incipient lateral tubercle is present already in the first postaxial vertebra, becoming a true synapophysis starting from the second postaxial. The tubercle and the synapophyses are followed by a depressed area similar to the one present in the axis, at least in the first presacrals for which this feature can be evaluated. Intercentra are consistently present between all presacral vertebrae that are exposed. These are more massive and rounded in the anterior part of the presacral section of the vertebral column (i.e., the cervical region; Fig. 6), but strip-like in ventral view in the trunk region, resembling ossified intervertebral discs (Fig. 7). The large and rounded third intercentrum has distinct posterolateral projections by the sides. Smaller projections are also present in the fourth and maybe even the fifth intercentrum. According to *Cocude-Michel (1963)* and *Fabre (1981)*, free presacral intercentra are limited to the cervical region in *Homoeosaurus* and *Kallimodon*, but present in the dorsal region as well in *Sapheosaurus* and *Pleurosaurus*. *Vadasaurus* lacks free presacral intercentra (*Bever & Norell, 2017*) and *Cocude-Michel (1967b)* mentioned a complete absence of free postcervical intercentra in the Teyler Museum specimen of *H. maximiliani*. *Ankylosphenodon* lacks intercentra at least in the thoracolumbar region, but this feature cannot be evaluated in the

rest of the vertebral column (*Reynoso, 2000*). Intercentra are consistently present in the vertebral column of *Sphenodon* (*Hoffstetter & Gasc, 1969*; *Fabre, 1981*), *C. hudsoni* (*Fraser, 1988*), and *Planocephalosaurus* (*Fraser & Walkden, 1984*).

The sacral vertebrae (Fig. 8) are mostly covered by bones of the pelvic girdle, but the exposed portion displays a centrum morphology that is similar to that of the presacrals. The exposed left transverse process (including the sacral rib) of the first sacral is strongly constricted close to its contact with the centrum and gradually and considerably expanded distally, with the distal portion assuming a fan-like shape in ventral view. The thinnest point occurs at around one fourth of the length of the process from its contact with the centrum. The distal end is more than five times wider than the thinnest point (3.1 mm vs 0.6 mm). This morphology clearly differs from the more cylindrical process of the first sacral in *Homoeosaurus*, *Kallimodon*, *Pleurosaurus* (*Cocude-Michel, 1963*), *C. hudsoni* (*Fraser, 1988*), and the extant *Sphenodon* (*Hoffstetter & Gasc, 1969*; *Fabre, 1981*; A.Villa, 2019, personal observation). Transverse processes of the first sacral in *Sapheosaurus* (as figured by *Cocude-Michel, 1963*: fig. 17B, and *Fabre, 1981*: fig. 46), *Piocormus* (based on drawings and figures by *Fabre, 1981*), and *Ankylosphenodon* (see *Reynoso, 2000*: fig. 5) seem to approach more the condition displayed by SNSB-BSPG 1993 XVIII 4, even though the difference in width between the proximal and distal ends is not as extreme. The second sacral has more homogenous, elongate transverse processes, which are less narrow close to the base and less expanded at the distal end. At the centrum, the process is equal in width to the latter, but moving laterally it loses a bit of width. The right transverse process of this vertebra is either largely missing or not exposed, whereas the better-preserved left one shows some damage to its posterior margin. In spite of this, the base of the posterior process is visible on both sides; the processes were therefore bifurcated (like other fossil forms, but unlike the extant *Sphenodon*; *Hoffstetter & Gasc, 1969*), even though a description of the morphology of the posterior process is not possible. Based on the preserved portion, it can be assumed that it was small, perhaps similar to the shape of the posterior process of *Youngina* (*Gow, 1975*). The posterior process originates above the base of the rib, similar to e.g., *Pleurosaurus* and unlike e.g., *Vadasaurus* and at least some specimens of *Kallimodon*. Distally, the anterior section of the transverse process curves smoothly about 30° towards the anterior, ending abruptly in a broad facet. As clearly visible on the left side, sacral transverse processes contact each other laterally. Strip-like intercentra are present both between the two sacrals and between the second sacral and the first caudal vertebra.

The first caudals (Fig. 8) are similar to the trunk vertebrae in the morphology of their centra, but then become more elongated. An autotomy plane is seen starting from the seventh caudal at the midlength of the vertebra. The first autotomic vertebra is located more anterior in the tail compared with *Sphenodon* (*Hoffstetter & Gasc, 1969*), *Kallimodon* (*Cocude-Michel, 1963*), *Ankylosphenodon* (if autotomy is actually present in this taxon; *Reynoso, 2000*), and possibly *Vadasaurus* (*Bever & Norell, 2017*). Autotomy may start even more anteriorly in *Sapheosaurus*, but this cannot be stated with complete confidence based on the available material (*Cocude-Michel, 1963*). In contrast, *Pleurosaurus* has no autotomic planes in the tail (*Cocude-Michel, 1963*; *Fabre, 1981*). Well-developed transverse

processes are present in caudal vertebrae 1 to 7. Unlike the first six caudals, which are exposed in ventral view, the seventh caudal is exposed in lateral view, and thus the transverse process is broken off and displaced dorsally. The process is similar in shape to the ones of the preceding vertebrae, but only about half as long as in the sixth vertebra. From the eighth caudal onwards (Figs. 8, 9), the transverse processes seem to be developed only as small lateral bumps, which disappear in more distal caudals. In the first six caudals, the transverse processes are elongated processes, which narrow distally. They are very well developed in the first caudal and then decrease in development posteriorly. All of them bend anterolaterally and this becomes even more pronounced posteriorly. Only *Sphenodon* has these markedly anterolaterally pointing transverse processes, but they start slightly more posterior in the caudal series, as the first few transverse processes are oriented strictly laterally in this taxon. On the contrary, *H. maximiliani*, *Kallimodon*, *Derasmosaurus*, *Oenosaurus*, *Piocormus*, *Vadasaurus*, and maybe pleurosaurs have posteriorly-bent processes in the first caudal vertebrae. Some of the caudal vertebrae (posterior to the non-autotomic ones) are exposed ventrolaterally and show the narrow and elongated neural spine located at the posterior end of the dorsal surface of the neural arch. Between the first and the second caudal vertebrae, a strip-like intercentrum is present (Fig. 8). Thus, only two postpelvic intercentra are present, contra seven in sapheosaurs (*Fabre, 1981*). In *C. hudsoni*, a third postpelvic intercentrum is present between the second and the third caudal vertebra (*Fraser, 1988*), which is the case in *Sphenodon* as well (*Hoffstetter & Gasc, 1969*; A.Villa, 2019, personal observation). Subsequent vertebrae of SNSB-BSPG 1993 XVIII 4 display a chevron bone (Fig. 8). The first chevron in the tail of *Sphenofontis* is broken. The following two chevrons show slightly better preservation. The chevrons are Y-shaped and extend posteroventrally. They are dorsally closed until roughly the 11th caudal. The anterodorsal margin is concave and articulates mostly with the posteroventral margin of the preceding caudal. The dorsolateral corners are rather pointed, not rounded. Where the two arms of the Y-shape meet ventrally, the chevrons thicken slightly mediolaterally. The size of the chevrons decreases further caudally. They are present all the way up to the regenerated part of the tail.

The thoracic ribs (Fig. 7) are long and thin, with a furrow running along their lengths, creating hourglass-shaped cross-sections. The ribs become shorter closer to the pelvic girdle, and while the anterior ribs are generally angled posteriorly, the last ribs anterior to the pelvis are angled anteriorly in their proximal portions. Their proximal ends are widened into a single articular surface contacting the synapophyses of the related vertebra. Distally, the ribs again widen slightly before terminating convexly. Very thin gastralia are present (Fig. 7), but highly displaced and poorly preserved. An osteoderm cover is lacking, in contrast with *Pamizinsaurus* (*Reynoso, 1997*).

**Pectoral girdle and forelimb.** A slight degree of displacement is evident in the pectoral area (Fig. 10). The interclavicle is largely covered by other bones, only the anterior end and the posterior tip being visible. This bone is T-shaped. The anterior end bears two slender and rather short lateral processes, less than half as long as the posterior process. These are straight, projecting at 90° from the base, and not slightly posteriorly curved, as

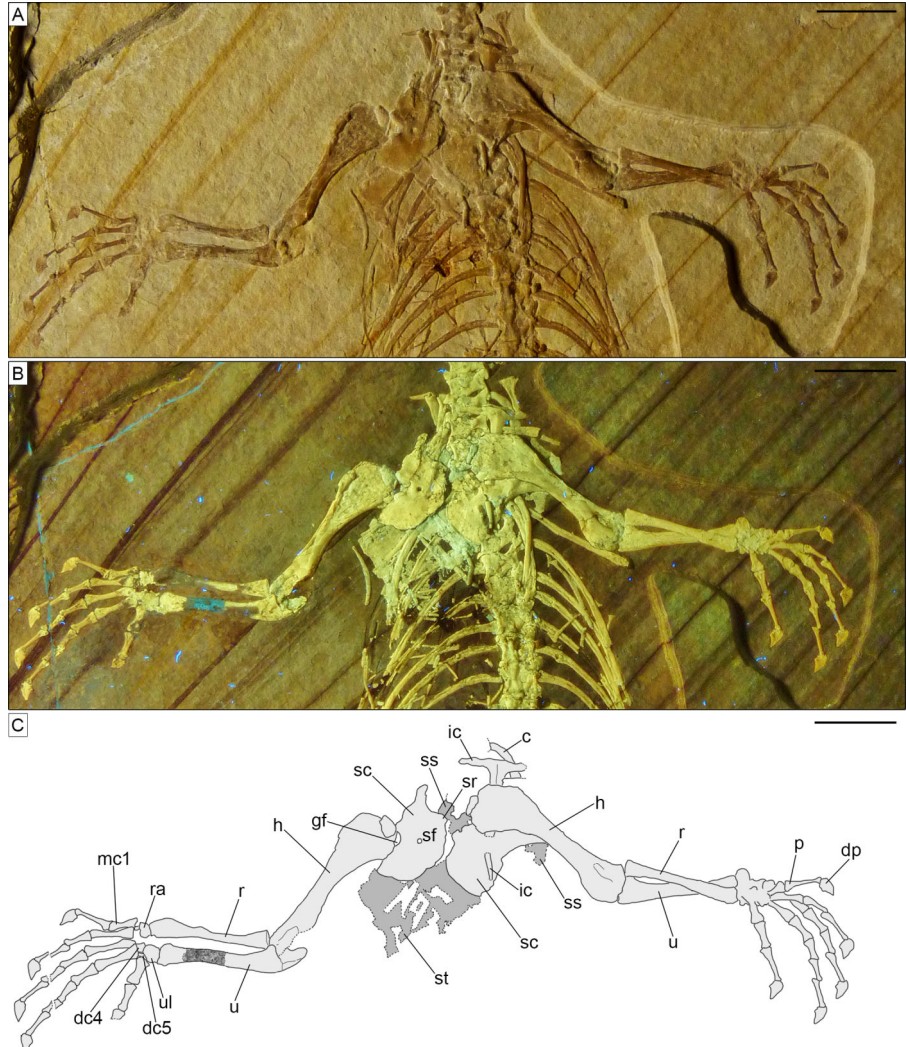

**Figure 10 Pectoral girdle and forelimbs of *Sphenofontis velserae* gen. et sp. nov.** (A) Standard light; (B) UV-light; interpretative drawing. Elements in plain dark grey are calcified, whereas patterned dark grey indicates reconstructed portions of bones. Scale bars = 1 cm. Abbreviations; c, clavicle; dc4-5, distal carpals 4 and 5; dp, distal phalanx; gf, glenoid fossa; h, humerus; ic, interclavicle; mc1, metacarpal 1; p, phalanx; r, radius; ra, radiale; sc, scapulocoracoid; sf, supracoracoid foramen; sr, scapular ray; ss, suprascapula; st, sternum; u, ulna; ul, ulnare. The UV-light photo in B was taken by Helmut Tischlinger.

reported by *Fabre (1981)* for *Pleurosaurus ginsburgi*. The anterior margin, although appearing relatively straight, contains a concavity on each of the lateral processes, lined by a small flange pointing ventrally on which the clavicles sat. The posterior margin of each lateral process is convex. The lateral ends of the processes appear rounded, not pointed. The center of the anterior margin of the interclavicle is very slightly concave, but not as much as sometimes seen in other rhynchocephalians. Whether the clavicles came into contact is unknown, but a middle anterior ridge as in *Gephyrosaurus* (*Evans, 1981*), or a real anterior process, is not present. The long posterior process narrows posteriorly, ending with an almost pointed tip. The posteriormost piece is thinner and

round in cross section. The ventral surface of the interclavicle has a median ridge formed by the confluence of the gently sloping sides. The ridge runs anteroposteriorly on the ventral surface along the main axis of the interclavicle, becoming less pronounced (but still visible) posteriorly. A similar ridge is seen in *Priosphenodon avelasi* (*Apesteguía, 2008*). The transition into the lateral processes is rounded, but does not have the "wing-like" coracoid facets that are seen in *P. avelasi* (*Apesteguía & Novas, 2003*; *Apesteguía, 2008*).

A probable clavicle is seen lying next to the 5th vertebra, partially underneath the interclavicle. It has a similar thickness as the ribs, but does not have the furrow running along its length. It also curves slightly stronger in the proximal region.

Both scapulocoracoids are preserved, but only the right one is completely exposed. In these bones, scapulae and coracoids are completely fused. They are large and have a roughly semicircular shape in ventral view. Laterally, the glenoid fossa is visible as a small notch, with a distinct superior buttress. The scapular contribution to the glenoid fossa appears larger than the coracoid contribution. Both the glenoid facets on the coracoid and scapular portions are significantly raised, the scapular one slightly more so. The supracoracoid foramen is visible just anteromedial to the fossa, roughly in the middle of the scapulocoracoid. The medial margin of the coracoid portion has no fenestration: it is convex, but becomes relatively straight where the coracoid contacts the sternum. The posterior part of the coracoid is elongate; the posteromedial margin is convex, but the posterolateral margin is slightly concave adjacent to glenoid facet. A similar shape of the posterior half of the coracoid is seen in the extant *Sphenodon* (*Howes & Swinnerton, 1901*). The scapular portion is an elongated and straight expansion, which is, however, poorly preserved in the right scapulocoracoid and almost completely covered by the humerus on the left side. It is posteriorly concave and its anterior margin cannot be seen. A very short and moderately wide scapular ray is present; it is separated from the main body of the scapula by a wide and shallow notch for the scapular fenestra and from the coracoid by a very shallow notch for the scapulocoracoid fenestra. This condition is reminiscent of what is seen in *Planocephalosaurus* (*Fraser & Walkden, 1984*), even though the latter taxon has a deeper notch for the scapular fenestra and no notch for the scapulocoracoid fenestra. Based on the CT scan of a single left scapulocoracoid figured by *O'Brien, Whiteside & Marshall (2018)*, it is not clear whether a morphology more or less similar to that of *Planocephalosaurus* could be shared by at least *C. hudsoni* as well or not. It has to be noted, however, that *Fraser (1988)* mentioned a *Sphenodon* specimen showing incipient scapular fenestration similar to that of *Planocephalosaurus*, thus suggesting that this condition might be present as a variable feature in other rhynchocephalians as well. This seems to be confirmed by our personal observations on CT data of extant *Sphenodon* (R. Montie, 2020, personal observation).

Large sheets of poorly ossified bones largely covered by the scapulocoracoid of SNSB-BSPG 1993 XVIII 4 on the right side and by the humerus on the left side probably represent the suprascapulae. Another skeletal element visible medial to, and in contact with, the scapulocoracoids is likely the sternum, which, based on its preservation, seems to have been largely cartilaginous. This element is a poorly preserved wide sheet, probably representing the presternum.
The humeri are quite long relative to the presacral vertebral column, with a slender shaft that strongly expands at the ends. Even considering compression, the minimal width of the shaft is less than half the maximal width of the ends. Nevertheless, the humeri are no less robust than in most other rhynchocephalians. Both humeri are exposed in ventral view. The anterior outline of the humerus is relatively straight, whereas the posterior one is distinctly concave. The proximal epiphysis is very wide, even though this large width could have been slightly enhanced by taphonomic compression; it displays a wide and moderately deep bicipital fossa. Only around midshaft does the concavity of the fossa disappear. Both the medial and lateral tuberosities appear small and poorly individualized. On the ventral surface of the latter, the deltopectoral crest is moderately developed. The humeral crest is also moderately developed. The line connecting the lateral tuberosity and the humeral condyle is straight and slightly oblique in ventral view. A small ossified plate caps the humeral condyle on both humeri, not being fused with the latter and possibly representing articular cartilage. Only a very slight twisting appears to be present on the humeri, unlike the 90° twisting of the humeri of *Sphenodon*. The distal epiphysis is wider than the shaft, but narrower than the proximal epiphysis. The left one is better preserved than the right one. A narrow but rather deep radioulnar fossa is visible, as is the entepicondylar foramen. The entepicondyle is robust, but poorly projecting. Because of this, the margin connecting the entepicondyle to the shaft is rather straight compared to the main axis of the humerus. In any case, the entepicondyle is still much more expanded than the ectepicondyle, thus resulting in the concave posterior outline of the humerus. Indeed, the ectepicondyle appears to hardly expand at all. Although a larger entepicondyle is quite common in rhynchocephalians (e.g., *Clevosaurus*, *Derasmosaurus*, *Gephyrosaurus*, *Kallimodon*; *Cocude-Michel, 1963*; *Evans, 1981*; *Barbera & Macuglia, 1988*; *Fraser, 1988*; *O'Brien, Whiteside & Marshall, 2018*), there are also some taxa that have an almost equally large ectepicondyle (e.g., *Ankylosphenodon*; *Reynoso, 2000*; *Sphenodon* and *Oenosaurus*; R. Montie, 2020, personal observation). The distal portion of the epiphysis appears well ossified, but it is poorly preserved. A small, cylindrical radial condyle is distinguishable on the right humerus.

Considering their overall width in relation to their total length, ulna and radius can be described as long and slender bones, with the ulna being slightly more robust. In both bones, the epiphyses are slightly expanded compared to the shafts and well ossified. Their proximal epiphyses are both curved slightly anteriorly. The proximal epiphysis of the ulna hosts a concave surface, the sigmoid (or trochlear) notch, for the articulation with the ulnar condyle (trochlea) of the humerus. Because of the displacement, however, the epiphysis seems to contact the radial condyle on the right side of the specimen. The olecranon process, which is exposed (even though poorly preserved) only on the right side, is well ossified but not fused to the rest of the ulna. The distal epiphyses of both radius and ulna are quite rounded.

The carpus is poorly preserved and probably poorly ossified (judged by the granular bone surface) on both sides. Nevertheless, a large and squared ulnare, a possible elongated radiale, and (only in the right manus) at least a relatively large distal carpal 4 and a small distal carpal 5 are recognizable. The rest of the manus includes elongated and slender

metacarpals and phalanges. The length of the metacarpals is maximal in metacarpal 3 and minimal in metacarpal 1, with the latter being slightly more than half as long as the former. Metacarpals 2 and 4 are slightly shorter than metacarpal 3, whereas metacarpal 5 is only very slightly longer than metacarpal 1. Metacarpal 5 is also more robust than the other metacarpals. Metacarpal 1 does not show the enlarged proximal end that is observed in pleurosaurids (*Cocude-Michel, 1963*; *Bever & Norell, 2017*). Similarly, the entire first digit is not as robust as in *Ankylosphenodon* (*Reynoso, 2000*). Penultimate phalanges are all very similar to each other, but they are longer and thinner than the preceding phalanges, with a bilobed distal end and an expanded proximal base. The first phalanx becomes progressively more robust, but also shorter, the more phalanges the finger has. This is true for all but digit V, which has a relatively robust first phalanx. The articulating condyles of the phalanges can be seen in the left manus, in which each phalanx distal to the most proximal one has a clear proximal condyle, which sockets into a notch on the preceding phalanx. These condyles have a slight U shape when seen from the proximal side.
The ungual phalanges are short and triangular in lateral or medial view, differing from the squared shape they have in *P. avelasi* (*Apesteguía & Novas, 2003*). They look similar on all digits, with no real morphological or size differences between them. They are very high and very short, with the length never exceeding twice their maximal height. The ventral flexor tubercle is large. The articulating surface of the distal phalanx with the penultimate phalanx is concave. The tips of the claw-like distal phalanges are very sharp. The phalangeal formula is 2-3-4-5-3. In the right manus, digit V seems to have one phalanx less, but this is due to a breakage at the level of the proximal epiphysis of the second phalanx.

**Pelvic girdle and hindlimb.** Elements of the pelvic girdles (Fig. 11) are not fused to each other. They are all very wide, in contrast to the more slender elements seen in *Kallimodon*. Both ilia are poorly visible in ventral view. These bones are anteroposteriorly elongated and rather slender (based on what can be seen, maximal height seems less than one third of the length). Anteriorly, a long expansion capped the pubis. The ilium seems to contribute most to the formation of the wide acetabulum, the concavity of which can be seen just dorsal to the ischium facet. The acetabular concavity continues through the ilioischiadic junction, however, which implies that at least a part of the acetabulum was formed by the ischium. On the better-preserved left side, the posterior (or dorsal) process of the ilium cannot be observed in its full length as it is partially covered by the left femur, but it appears to reach just past the second sacral transverse process.

The left pubis is moderately preserved and still in contact with the ilium, in contrast with the very poor preservation of the fragmentary right element. The symphyseal portion of the pubis is anteroposteriorly wide in ventral view. The symphysial margin is not significantly expanded anteroposteriorly, and as such the symphyseal portion is not hourglass-shaped, as is the case in many other taxa from the Solnhofen Archipelago (e.g., *Homoeosaurus*, *Kallimodon*). The anterior margin of the symphyseal process is very slightly concave; almost straight. On the anterolateral side of the pubis there is a short and wide processus lateralis pubis, hosting a distinct pubic tubercle on its top. Despite the

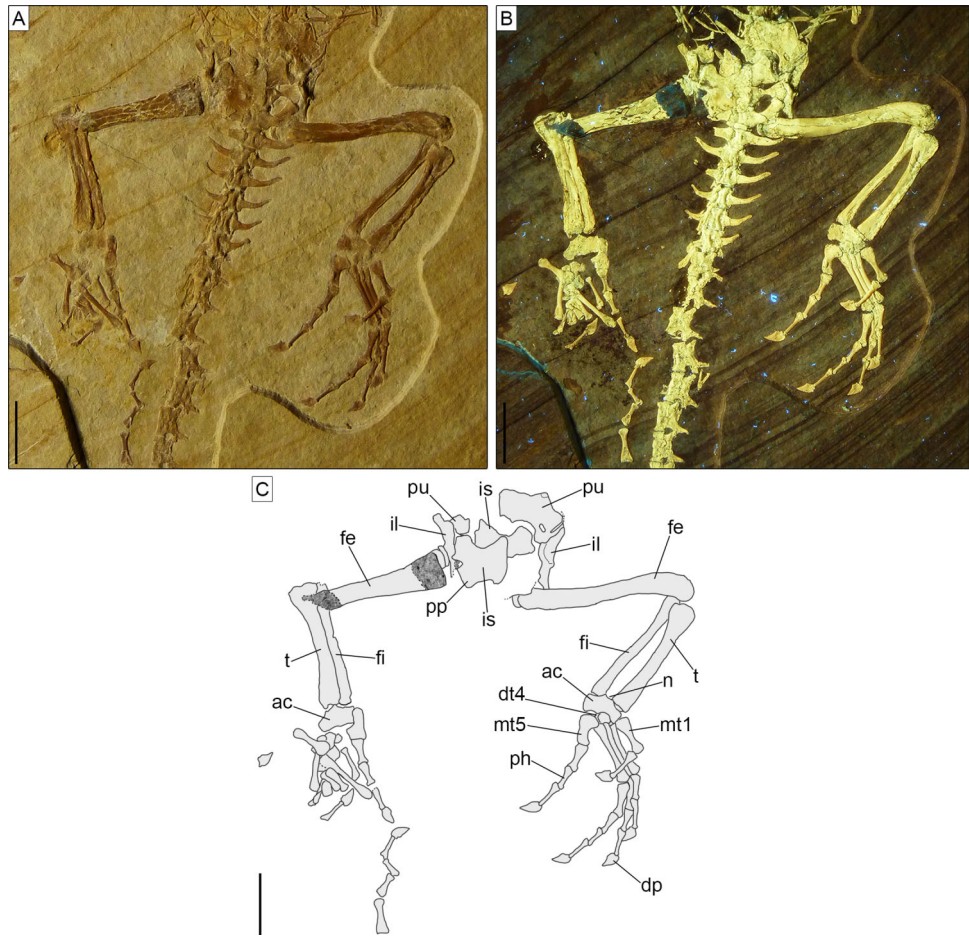

**Figure 11 Pelvic girdle and hindlimbs of *Sphenofontis velserae* gen. et sp. nov.** (A) Standard light; (B) UV-light; (C) interpretative drawing. Patterned dark grey indicates reconstructed portions of bones. Scale bars = 1 cm. Abbreviations: ac, astragalocalcaneum; dp, distal phalanx; dt4, distal tarsal 4; fe, femur; fi, fibula; il, ilium; is, ischium; mt1, metatarsal 1; mt5, metatarsal 5; n, proximal notch; ph, phalanx; pp, posterior process of the ischium; pu, pubis; t, tibia. The UV-light photo in B was taken by Helmut Tischlinger.

overall shortness of this process, the tubercle itself is clearly set off from the main body of the pubis. A small, anteroposteriorly-directed ridge leads up towards it, but this ridge likely represents the line along which the symphyseal portion of the pubis flexes medially. Lateral to the processus lateralis pubis the margin of the pubis is concave, as in *H. maximiliani*, *Sphenodon*, *P. avelasi*, and *Gephyrosaurus*, not convex, as seen in e.g., *Kallimodon pulchellus*, *Sapheosaurus*, and *Pleurosaurus*. In *Kallimodon*, *Pleurosaurus*, *Vadasaraurus*, and some specimens of *Sphenodon*, the tubercle, the ischium facet, and the obturator foramen are roughly aligned. A very wide obturator foramen is placed close to the suture with the ilium and ischium. A similar position is seen in *C. hudsoni* and *Planocephalosaurus* (*Fraser & Walkden, 1984*; *Fraser, 1988*). The foramen is oval in shape and located far posterior to the midline of the symphyseal process, lateral to the thyroid fenestra. The posterior margin of the pubis is strongly concave. Proximally, the contact surface with the rest of the girdle elements is almost completely occupied by the contact

with the ilium, whereas the ischium facet is quite small. The ilium appears to extend over the pubis as far as the apex of the lateral convexity of the head of the pubis. The proximal half of the pubis extends much further posteriorly than it does anteriorly. The pubis contributes at least 50% to the thyroid fenestra.

The right ischium is rather well preserved and exposed, largely covering the left one. It is anteroposteriorly very wide and rather short, with its maximal width being around two thirds of the length. It has a deeply concave anterior margin, due to distinct anterior extensions of both the proximal and the distal ends. This margin defines the posterior border of the thyroid fenestra. The articular facet with the pubis is smaller, about half the size of that with the ilium. The latter is slightly concave. The posterior margin is damaged, but the base of a wide posterior process is visible. The posterior margin of the ischium shows a shallow concavity distal to the posterior process, again similar to *Sphenodon*, and unlike the deep concavities seen in e.g., *Kallimodon*, or the convex margins of e.g., *Youngina* and *Gephyrosaurus* (*Gow, 1975*; *Evans, 1981*). The distal end of the ischium is almost twice as wide as its proximal end.

The femora are long and slender, with well-ossified epiphyses and a slightly sigmoid shape with a small degree of torsion. On the left femur, the femoral condyle articulating with the acetabulum can clearly be seen jutting out proximally. The femoral condyle is large and robust, with a ridge that disappears about halfway distally on the shaft of the femur. The distal end of the femur is also widened and rounded in distal outline. The exposed anterior condyle is robust. The femur of SNSB-BSPG 1993 XVIII 4 is longer relative to the presacral vertebral column than that of any other known rhynchocephalian.

Tibiae and fibulae are also long, slender, and well ossified. They are similar in length, although the former is slightly more robust than the latter. They are both shorter and narrower than the femur. Moreover, the expansion of the epiphyses compared to the shaft is stronger in the tibia than in the fibula. The fibula is very rod-like, with only small proximal and distal expansions. The proximal expansion of the tibia is much more pronounced. Based on what can be observed on the left hindlimb of the specimen, which is better preserved and thus more closely approaching the original condition in the living animal, the distal heads of the tibia and fibula do not come into contact with each other at the articulation with the pes.

The pes is better preserved on the left side. Astragalus and calcaneum are fused. In the mediolaterally elongated astragalocalcaneum, the tibial and the fibular articular facets are separated by a rather wide and shallow proximal notch (not present in *Clevosaurus hudsoni*; *O'Brien, Whiteside & Marshall, 2018*). Only one distal tarsal, likely the large and subpentagonal distal tarsal 4, is visible. It has a clear notch on the distal side, which is oriented towards the middle three digits. Vague shapes of distal tarsals 1 to 3 can be seen, but it is unclear whether they are fused or not. Metatarsals and phalanges are long and slender. The length of the metatarsals is greatest in metatarsals 3 and 4. It decreases slightly in metatarsal 2 and distinctly in metatarsal 1. Metatarsals 2 and 1 are about 80% and 60% as long as metatarsals 3 and 4, respectively. Metatarsal 5 is very short. The robustness of these bones follows a reversed pattern, with a very robust metatarsal 5, a slightly robust metatarsal 1, and equally narrow metatarsals 2, 3, and 4. The shape of metatarsals 2, 3,

and 4 is exactly the same as that of metacarpals 2, 3, and 4, only quite a lot longer. Metatarsal 5 is hook-shaped, but not as acutely concave laterally as in *Kallimodon*. Its distal end is straight, not very expanded. Its proximal edge is convex and articulates with the astragalocalcaneum and distal tarsal 4. It displays a prominent tubercle on its ventral surface, close to its distal end. The morphology of the phalanges in the pes is generally equivalent to what is seen in the manus, except for an increase in robustness and (slightly) in length in the former. The first phalanx of digit IV is quite large. Digit I is not very much larger or much more robust than the other digits, something that is seen also in e.g., *Vadasaurus* and *Kallimodon pulchellus*. The phalangeal formula is 2-3-4-5-4.

### Remarks

A number of features support the recognition of SNSB-BSPG 1993 XVIII 4 as a subadult individual, which still had to reach fully-grown adulthood. Evidence supporting this assumption is found both in the skull and in the postcranium. First of all, the specimen displays a rather advanced degree of ossification, especially when considering the girdles and limbs. This is particularly evident in the epiphyses of the long bones, even though the lack of a complete fusion of the olecranon with the rest of the ulna (Fig. 10) is a signal that the growth process was still active when the animal died. Complete fusion of the astragalocalcaneum (Fig. 11), without any sign of a suture line, is also indicative of a rather late ontogenetic stage for SNSB-BSPG 1993 XVIII 4 (*Russell & Bauer, 2008*). The same holds true for the presence of a distinct processus lateralis pubis, which is absent in juvenile rhynchocephalians, according to *Fabre (1981)*. According to our personal observations on *Sphenodon*, the distal contact between the sacral transverse processes is also absent in early juveniles. Furthermore, the presence of caniniform successional teeth (Figs. 4, 5) may also be related to late ontogenetic stages (*Reynoso, 2003*; *Romo de Vivar et al., 2020*). The unfused exoccipitals and opisthotics (Fig. 3) are generally a juvenile character, but *Evans (2008)*: p. 72) stated that fusion in the adult is just possible and thus not always the case. *Jones et al. (2009a)* also figured two rather large (and thus presumably not at least early juvenile) skulls of *Sphenodon* with unfused exoccipitals and opisthotics. Three hypoglossal foramina are also a feature of post-hatchling individuals, even though fully-grown adults only display two (*Evans, 2008*). Finally, the premaxillae bear well-individualized teeth (Figs. 4, 5), still not coalescing into the chisel-like structure that is seen in older individuals in most rhynchocephalians.

## DISCUSSION

In their overview of the Brunn vertebrate fauna, *Rauhut et al. (2017)* already recognised the morphological peculiarities and the possible new taxonomic identity of SNSB-BSPG 1993 XVIII 4. We can herein confirm this, describing this specimen as a new taxon, *Sphenofontis velserae* gen. et sp. nov. This new taxon clearly displays features of derived rhynchocephalians (Eusphenodontia sensu *Herrera-Flores et al., 2018*), such as the incipient coalescence of the premaxillary teeth (likely leading to a chisel-like premaxillary structure in individuals older than the one represented by the holotype) and the reduced palatal dentition. Furthermore, it can be recognised as part of Neosphenodontia

(*Herrera-Flores et al., 2018*) due to the following characters: a single row of palatine teeth; no pterygoid teeth; presence of a posterior process of the ischium. The presence of a caniniform tooth following an edentulous gap was proposed by *Reynoso (1996, 2003)* to diagnose sphenodontine sphenodontids. This suggests that *Sphenofontis* can also be referred to this clade, even though it should be noted that more investigation is needed to understand the real taxonomic significance of caniniform successional teeth in rhynchocephalians (*Apesteguía, Gómez & Rougier, 2012*). Nevertheless, comparisons with other rhynchocephalian taxa (see Description above) highlight a strong morphological resemblance between *Sphenofontis* and other sphenodontines, and *Sphenodon* in particular. This further supports the sphenodontine identity of the Brunn taxon. The skull of *Sphenofontis* recalls the extant *Sphenodon* in morphological features of e.g., the jugal, the postfrontal/postorbital joint, the quadrate, the squamosal, the basioccipital, and the prootics. Other features are shared with representatives of more early-branching clades, though, including the overall skull shape (shared with *Homeosaurus* and clevosaurids), the proportions of the premaxillary body (shared with *Planocephalosaurus*, but also with the eilenodontine *Sphenotitan*), and the presence of a posterodorsal process of the premaxilla (shared with *Clevosaurus*). If the identification of *Sphenofontis* as a sphenodontine is correct, this mixture of characters may suggest a basal position within the clade. A preliminary phylogenetic analysis based on the matrix recently published by *Simões, Caldwell & Pierce (2020)* and conducted as a first test for the relationships of *Sphenofontis* supports this conclusion (Fig. 12; see the Data S1 for further details).

The heterodont premaxillary dentition of SNSB-BSPG 1993 XVIII 4 (Figs. 4, 5) also strongly resembles that of a specimen of *Sphenodon punctatus* used for comparison, SNSB-BSPG 1954 I 454. Like in the Jurassic fossil, this specimen shows three premaxillary teeth, including a large and slightly more isolated lateral one and two smaller medial teeth. In contrast to the situation in the fossil taxon, all three teeth are coalesced at their bases, the mesial two teeth more so than the lateral one. In contrast with *Sphenofontis*, in which the mesialmost tooth is the smallest, in SNSB-BSPG 1954 I 454 the most mesial tooth is significantly larger than the second premaxillary tooth. Flanges on the premaxillary teeth of SNSB-BSPG 1954 I 454 show the same pattern as in *Sphenofontis*, but it is not possible to evaluate the presence of lingual striae, due to strong wear of this side in the largest premaxillary teeth. In *Sphenodon*, multiple teeth present in each premaxilla in the hatchling end up with complete fusion into a single chisel-like structure with increasing age (*Robinson, 1976*; *Evans, 2008*; *Jones et al., 2009a*). This happens in fossil rhynchocephalians as well: in *Vadasaurus*, for example, the single premaxillary chisel-like structure apparently originated from the fusion of three incisiform teeth (*Bever & Norell, 2017*), whereas two teeth fuse to form a single structure in adult *Homoeosaurus maximiliani* and *Kallimodon*, according to *Fabre (1981)*, and in *Brachyrhinodon*, according to *Fraser & Benton (1989)*. *Clevosaurus hudsoni* and *Clevosaurus convallis* have either three or four premaxillary teeth, with the most lateral one being larger than the others at least in the former species (*Fraser, 1988*; *Säilä, 2005*; *Hsiou, De França & Ferigolo, 2015*). *Clevosaurus minor* only has three, equally-sized premaxillary teeth (*Fraser, 1988*), whereas fossils referred to *C. brasiliensis*, *C. bairdi*, and Chinese *Clevosaurus* show a

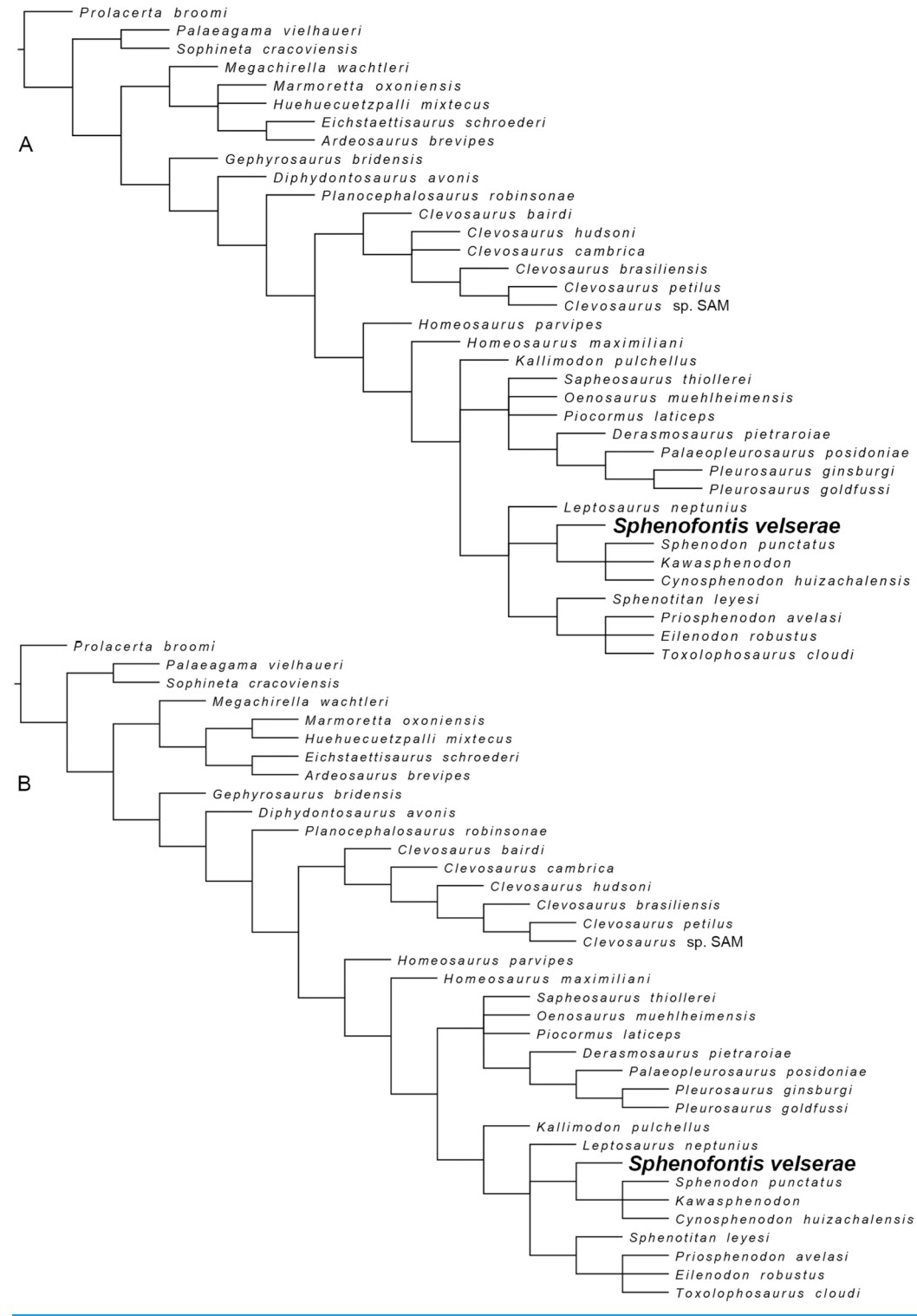

**Figure 12 Phylogenetic relationships of *Sphenofontis velserae* gen. et sp. nov.** (A) Strict consensus tree under equal weights maximum parsimony; (B) Strict consensus tree under implied weighting maximum parsimony.
single, tusk-like premaxillary "incisor" (*Sues, Shubin & Olsen, 1994*; *Hsiou, De França & Ferigolo, 2015*; but note that *Jones, 2006*, mentioned the presence of two or three cusps in the chisel-like structure of at least one of the Chinese specimens). An ontogenetic shift from multiple distinct teeth to a single chisel-like cutting edge is seen in *Clevosaurus* as well, at least based on what can be observed in *C. hudsoni*, *C. minor*, and *C. convallis* (*Fraser, 1988*; *Säilä, 2005*); the single "incisor" seen in some taxa may therefore just reflect their older age. *Planocephalosaurus*, on the other hand, has four premaxillary teeth that remain individualized throughout ontogeny (*Fraser, 1982*), whereas a single chisel structure is found in both small and large individuals (juveniles and adults?) of *Sphenotitan* (*Martínez et al., 2013*). Despite these latter taxa, variation in premaxillary tooth count between different fossil rhynchocephalians may therefore be just due to different ontogenetic stages or to simple individual variation. Nevertheless, *Cocude-Michel (1963)* counted two morphologically-similar premaxillary teeth in *Homoeosaurus maximiliani*, one in *Pleurosaurus*, and either one or two in *Kallimodon*. *Fabre (1981)* mentioned only two coalescing premaxillary teeth in *Sphenodon*, based on the specimen available to him to study, and considered the presence of two well-differentiated (but coalescing at the base) teeth in each premaxilla of *Homoeosaurus maximiliani* as a juvenile character. *Fabre (1981)* observed a similar condition in the type of *Leptosaurus neptunius*. All known premaxillae of *Rebbanasaurus* and the only known (post-hatchling) specimen of *Pamizinsaurus* display three teeth (*Reynoso, 1997*; *Evans, Prasad & Manhas, 2001*), which increase in size from medial to lateral, whereas four teeth are present in the single premaxilla attributed to *Godavarisaurus* (*Evans, Prasad & Manhas, 2001*). The single premaxilla attributed to *Fraserosphenodon* (*Fraser, 1993*; *Herrera-Flores et al., 2018*; referred to *Clevosaurus* sp. by *Fraser, 1988*) is distinctly different from SNSB-BSPG 1993 XVIII 4 in having two large teeth followed laterally by a markedly smaller third tooth; the two largest teeth are partially coalescing, thus suggesting a developmental pattern similar to other rhynchocephalians (*Herrera-Flores et al., 2018*). *Polysphenodon* probably had two premaxillary teeth (*Fraser & Benton, 1989*), as is the case for the single premaxilla tentatively referred to *Cynosphenodon* by *Reynoso (1996)*. Apart from *Planocephalosaurus*, four premaxillary teeth are also present in a small sphenodontian from the Kimmeridgian of Schamhaupten that was originally referred to *Leptosaurus* (*Renesto & Viohl, 1997*; see also *Rauhut & López-Arbarello, 2016*).

When considered as a whole, the distinct and peculiar heterodont dentition shown by SNSB-BSPG 1993 XVIII 4 (Figs. 4, 5) is not seen in any other fossil rhynchocephalian. Of course, this is based on a single and not-completely mature specimen, and some degree of variation may be expected if further material referable to the same taxon becomes available. However, the observed features are still notable. This is particularly true for the complex size trend in the additional dentition on the maxillae, as well as for the coalescing teeth followed by an isolated, canine-like third tooth visible in both the premaxilla and the anterior end of the dentary, even though the latter may at least in part be influenced by ontogenetic variation. As far as the former feature is concerned, particularly interesting, and likely significant, is the very small size and medial displacement of the fourth maxillary tooth. An aberrant nature of this tooth may be considered, especially given that only a

single specimen of the new taxon is known and thus the feature cannot be confirmed by other individuals. However, we are aware of no other case of tooth displacement such as the one shown by SNSB-BSPG 1993 XVIII 4 in any acrodont lepidosaur (*Rothschild, Schultze & Pellegrini, 2012*), and the symmetrical condition visible in this specimen hints against a pathological origin. *Cynosphenodon* displays a very small tooth (denticle sensu *Reynoso, 1996*) in the middle of the additional series as well, but this was described for the dentary in this taxon (unknown in the maxilla; *Reynoso, 1996*). As clearly shown in our description, this feature is only present in the maxilla in the Brunn taxon. It has to be noted that *Cynosphenodon* also has an alternating size pattern in the maxillary hatchling dentition (*Reynoso, 1996*: fig. 6B), but the successional dentition is unknown in this Mexican taxon and the hatchling dentition is heavily worn in the German specimen, thus precluding a comparison of the tooth-size trends in the maxilla between them. Somehow comparably with SNSB-BSPG 1993 XVIII 4, *Sphenocondor* also has different-sized successional teeth on the dentary, with the posteriormost one larger than and clearly separated from those located anterior to it. However, successional dentary teeth of *Sphenocondor* differ from those of *Sphenofontis* in being strongly recurved and more notably striated (*Apesteguía, Gómez & Rougier, 2012*). Furthermore, the exact number of successional dentary teeth in *Sphenocondor* is unclear. In their description, *Apesteguía, Gómez & Rougier (2012)* mentioned two preserved teeth plus a possible third one. However, two is the number of these teeth reported in their tab. 2, noting also space for three "anterior" teeth. These missing teeth mentioned in the table are hypothesised based on the close relationship between *Sphenocondor* and *Godavarisaurus* found in *Apesteguía, Gómez & Rougier (2012)* phylogenetic analysis. Thus, a possible complete count of five successionals is hypothesised by the authors, as confirmed by their labelling of the posteriormost successional tooth as the fifth in their fig. 4 (even though they do not include the first tooth in their drawing, starting from the second one instead). In spite of this, they write in the text that the successional dentition of *Sphenocondor* encompasses "at least three teeth (probably four)" (*Apesteguía, Gómez & Rougier, 2012*: p. 346) and three successional teeth plus a possible, missing fourth one anteriorly are depicted in their fig. 2. In any case, the number of successional teeth would be higher in *Sphenocondor* than in the holotype of *Sphenofontis*. The presence of the labial groove that is considered autapomorphic of *Sphenocondor* by *Apesteguía, Gómez & Rougier (2012)* cannot be clearly evaluated for the German taxon. Posterior to the successional dentition, the dentary of *Sphenocondor* also displays a small diastema and a series of teeth, the size of which increases towards the posterior end (*Apesteguía, Gómez & Rougier, 2012*). In contrast with SNSB-BSPG 1993 XVIII 4, however, teeth of this taxon seem not to show a decreasing trend in size for the last few teeth in this series. Nevertheless, post-successional dentary teeth in *Sphenocondor* are unstriated, as in SNSB-BSPG 1993 XVIII 4. The dentary dentition of the Brunn specimen further differs from the recently-described *Lanceirosphenodon* (*Romo de Vivar et al., 2020*) because of the non-alternating size of the additional teeth in the latter taxon, which shows a gradual decreasing trend instead.

Among European Jurassic forms, the absence of striae and, at least in the maxillae, flanges in most of the teeth of SNSB-BSPG 1993 XVIII 4 differs from the condition

observed in *Homoeosaurus*, *Kallimodon*, *Leptosaurus*, and *Pleurosaurus* (*Cocude-Michel, 1963*, *1967a*, *1967b*; *Fabre, 1981*). *Vadasaurus*, *Sigmala*, and *Pelecymala* have flanged teeth as well, but the presence of striae cannot be evaluated based on the description and figures given by *Bever & Norell (2017)* and *Fraser (1986, 1988)*. Triassic *Clevosaurus* all possess flanged maxillary teeth (*Sues, Shubin & Olsen, 1994*; *Säilä, 2005*; *Hsiou, De França & Ferigolo, 2015*; *O'Brien, Whiteside & Marshall, 2018*). Maxillary teeth of *Pamizinsaurus* are strongly striated (*Reynoso, 1997*). Both flanges and striae are known also in *Planocephalosaurus* from the Triassic of England, *Rebbanasaurus* from the Jurassic of India, and the holotypic maxilla of the Cretaceous *Lamarquesaurus cabazai*, which therefore also differ from the Brunn specimen in this respect (*Fraser, 1982*; *Evans, Prasad & Manhas, 2001*; *Apesteguía & Rougier, 2007*). *Godavarisaurus* has flanged but unstriated maxillary teeth (*Evans, Prasad & Manhas, 2001*). The detailed morphology of the maxillary teeth of *Brachyrhinodon* cannot be evaluated for preservational reasons, but they have flanges, as is probably the case for those of *Polysphenodon* as well (*Fraser & Benton, 1989*). Teeth devoid of both flanges and striae are reported for the dentaries referred to cf. *Diphydontosaurus* sp. from the Triassic of Vellberg (*Jones et al., 2013*). Similar to SNSB-BSPG 1993 XVIII 4, a complex pattern of alternation in tooth size is also present in *Clevosaurus brasiliensis* and *C. minor* (see *Bonaparte & Sues, 2006*, *Hsiou, De França & Ferigolo, 2015*, and *Fraser, 1988*, respectively), even though the pattern is different in these taxa when compared to the Brunn species, and moreover they display flanges in at least some maxillary teeth. A consistent posterior increase in size in the dentition, with the largest tooth being the last one and no posterior flanges on the teeth, is present in the dentary of *Tingitana* (*Evans & Sigogneau-Russell, 1997*). The same taxon has small and large teeth alternating in the maxilla, the dentition of which further differs from that of SNSB-BSPG 1993 XVIII 4 because of the presence of posterior flanges. In addition to all of this, the dentition of SNSB-BSPG 1993 XVIII 4 does not show the opisthodontian condition of eilenodontine rhynchocephalians, typified by the absence of regionalization and the presence of a compact tooth row composed by mediolaterally-enlarged teeth (*Rasmussen & Callison, 1981*; *Apesteguía & Novas, 2003*; *Foster, 2003*; *Martínez et al., 2013*; *Apesteguía & Carballido, 2014*). Transversally broad teeth are also found in *Pelecymala* (*Fraser, 1986*) and *Fraserosphenodon* (*Fraser, 1993*; *Herrera-Flores et al., 2018*), thus representing a difference between SNSB-BSPG 1993 XVIII 4 and these Triassic genera. The posterior groove that is autapomorphic for *Kawasphenodon* (*Apesteguía, 2005*; *Apesteguía, Gómez & Rougier, 2014*) is also absent in the dentition of *Sphenofontis*, which further differs from the South American genus in having dentary teeth that are not squared at the base. *Deltadectes* has striated teeth provided with an apical longitudinal trough (*Whiteside, Duffin & Furrer, 2017*). Finally, both the very peculiar dentition of *Oenosaurus* (see *Rauhut et al., 2012*) and the continuously-growing, unregionalized teeth of *Ankylosphenodon* (*Reynoso, 2000*) are clearly different from the condition shown by *Sphenofontis*. The extant *Sphenodon* seems to show no striae on either maxillary or dentary teeth, whereas short flanges are present in at least some teeth in the maxillae and maybe also the dentaries (A.Villa, 2019, personal observation).

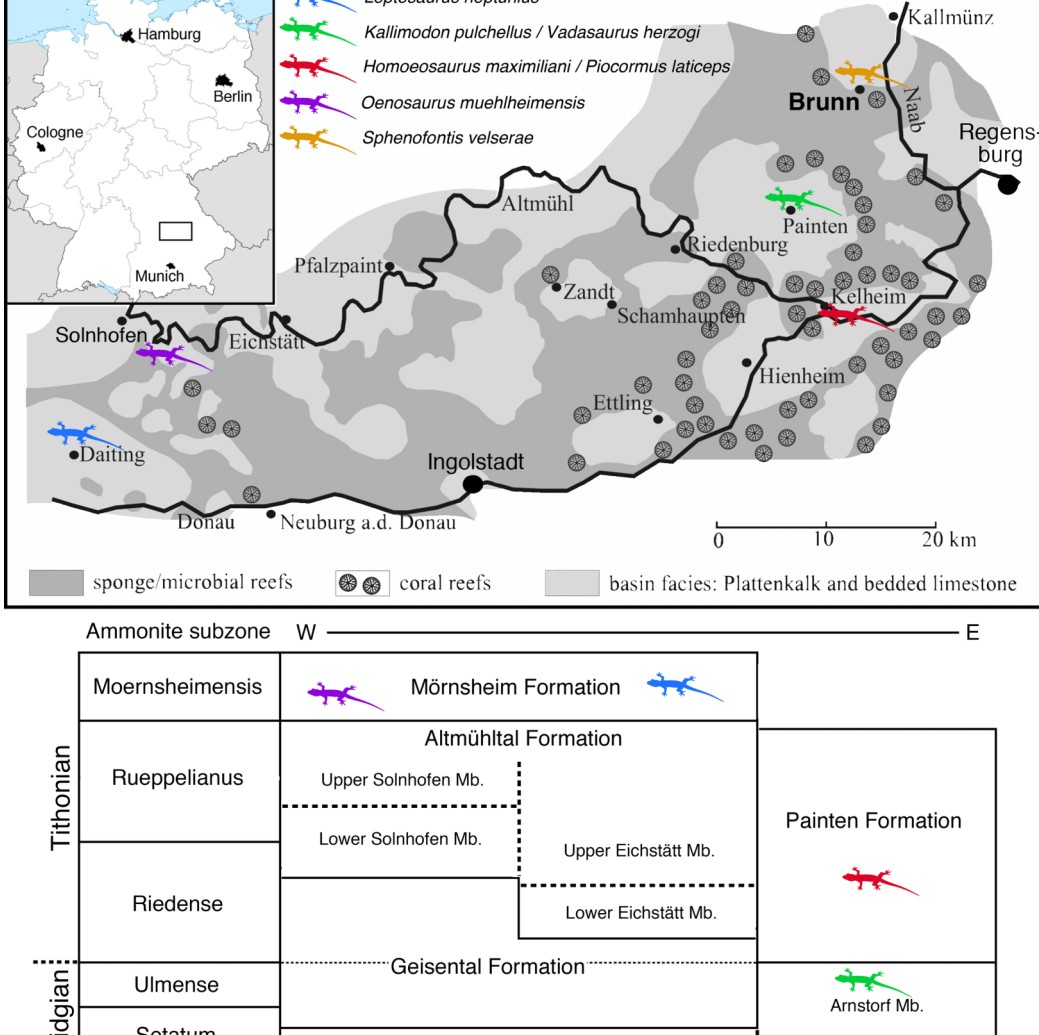

**Figure 13 Geographic and stratigraphic distribution of non-pleurosaurid rhynchocephalians in the Solnhofen Archipelago, based on the occurrences of the respective type specimens of the named taxa.** Map adapted from *Rauhut et al. (2017)*, stratigraphic scheme based on *Niebuhr & Pürner (2014)*.

Concerning the postcranial anatomy, *Sphenofontis* bears some similarities with *Sphenodon* as well (such as in the number of presacral vertebrae and in the orientation of the transverse processes in the anterior caudal vertebrae), but also with other extinct taxa, including non-sphenodontines. The persistence of intercentra in the whole presacral part of the vertebral column is a feature shared by a variety of rhynchocephalians, both within (*Pleurosaurus*, *Sapheosaurus*, *Sphenodon*) and outside (*Clevosaurus*, *Planocephalosaurus*) Neosphenodontia. It can therefore be interpreted as a plesiomorphic feature of the whole group, which was repeatedly lost in different clades (e.g., in *Ankylosphenodon*, *Homoeosaurus*, *Kallimodon*, and *Vadasaurus*). Other characters that

may be similarly interpreted are the proportional development of the entepicondyle and ectepicondyle of the humerus as well as the shape of the margin of the pubis lateral to its processus lateralis. Again, if *Sphenofontis* is indeed a sphenodontine, it shows the retention of possible plesiomorphic morphological features in its postcranium as well. The ratio of the length of the ulna to the length of the humerus (0.767) is similar to that of, among others, *Sphenodon* and *Gephyrosaurus*, but higher than in e.g., *Vadasaurus*, *Pleurosaurus*, *Sapheosaurus*, *Ankylosphenodon*, *Derasmosaurus*, and *Priosphenodon avelasi*. However, these proportions could be influenced by ecological habits and so their taxonomic significance need further study in order to be thoroughly understood. Of difficult interpretation is also the functional value of the peculiar morphology of the transverse process in the first sacral vertebra (Fig. 8). The shape observed in *Sphenofontis* is, to the best of our knowledge, not known in any other rhynchocephalian, or lepidosaurian reptiles in general. Thus, it might represent an autapomorphy of this Jurassic taxon. Its possible function, however, remains obscure for the moment. It may be somehow correlated with the anterolaterally-oriented transverse processes of the first caudal vertebra, which are also only known in this taxon among rhynchocephalians.

## CONCLUSIONS

Previous rhynchocephalian discoveries from the Late Jurassic limestones of southern Germany already proved the importance of the Solnhofen Archipelago to unravel the Mesozoic diversity of these reptiles, with at least six different genera represented in some cases by well-preserved, articulated specimens. *Sphenofontis velserae* gen. et sp. nov. adds to this diversity, with another specimen displaying an exquisite preservation that allows a detailed description of its morphology. Further morphological data will be potentially retrieved in the future with computed tomography scans, which were not available for our study but would certainly help to better understand significant osteological features of the taxon such as those located on the dorsal side of the cranium. *Sphenofontis* is here referred to Neosphenodontia and tentatively to Sphenodontinae, but it shows a combination of features that distinguish it from all other rhynchocephalians known so far, including some characters that may represent autapomorphies of the taxon. Our preliminary phylogenetic analysis supports *Sphenofontis* as an early-branching representative of the Sphenodontinae lineage. In the future, further scrutiny will permit a better understanding of its relationships with other rhynchocephalians, especially those from the Solnhofen Archipelago, and also to improve our comprehension of character distribution in less inclusive clades within the group due to its good preservation and the apparent mixture of derived and plesiomorphic features. This gains additional significance in the light of the importance of mosaic evolution in sphenodontian phylogenetic history recently highlighted by *Simões, Caldwell & Pierce (2020)*.

Given that the type locality of *Sphenofontis*, the Brunn quarry, represents the oldest part in the stratigraphic sequence of the Solnhofen Archipelago, the new taxon is one of the oldest rhynchocephalians from the area (Fig. 13), shedding some light on the earliest dispersal of these reptiles in the Archipelago. *Sphenofontis* supports the presence of less-morphologically-specialized rhynchocephalians in the early history of this area,

possibly already sharing its environment with forms related to taxa that would successively become more important in the terrestrial faunas of the islands though (two other specimens from Brunn may be related to *Kallimodon*; *Rauhut et al., 2017*). The new taxon does not display any evident specialization in its dentition, which was therefore most likely adapted to a generalist carnivorous/insectivorous diet comparable with that of the extant *Sphenodon* (*Lindsey & Morris, 2011*). The overall cranial and postcranial morphology lacks any clear adaptation towards an aquatic or semiaquatic mode of life, thus indicating that *Sphenofontis* thrived in the terrestrial ecosystems of the islands. Based on limb and body proportions, *Sphenofontis* agrees more with ground-dwelling than arboreal habits in having rather short limbs when compared with the presacral length (*Melville & Swain, 2000*). Furthermore, it shows no indication of running abilities in the relative length of the limb elements (i.e., there is no increase in zeugopodial elements compared to the stylopodial ones; *Miles, Fitzgerald & Snell, 1995*; *Li, Hsieh & Goldman, 2012*). Thus, a less-specialized ground-dwelling behavior may be suggested for this new taxon. Its precise mode of life needs further morphofunctional studies to be better understood, however.

Together with *Cynosphenodon* from Mexico, the new taxon demonstrates that taxa that are closely related and morphologically similar to the recent *Sphenodon* obviously already had a wide distribution in the Mid-Mesozoic, possibly testifying to the relictual status of the modern taxon.

## ACKNOWLEDGEMENTS

Lisa Velser found and prepared the specimen described here, and thus deserves our special thanks. We thank Winfried Werner for discussions about the geology of the Kimmeridgian-Tithonian limestones of the Franconian Alb and Oliver Voigt for help with the Leica microscope. The UV photos were taken by Helmut Tischlinger. Ilaria Paparella kindly shared photos of skeletonized specimen of *Sphenodon*, whereas Filippo Bertozzo discussed with us the possibility of a pathological nature of the strange maxillary tooth of *Sphenofontis*. We would also like to acknowledge the Academic Editor of PeerJ, Mark Young, and the three reviewers, Jorge Herrera Flores, Paulo de Vivar, and Nick Fraser, for helping improving our paper with their useful comments.

### Funding

This work was funded by the Alexander von Humboldt Foundation (Humboldt Research Fellowship to Andrea Villa) and by the Deutsche Forschungsgemeinschaft (Project RA 1012/28-1 to Oliver WM Rauhut). Excavations in Brunn were supported by the Bildungs- und Dokumentationszentrum Ostbayerische Erdgeschichte e.V. and the Freunde der Bayerischen Staatssammlung für Paläontologie und Geologie e.V. The funders had no role in study design, data collection and analysis, decision to publish, or preparation of the manuscript.

## Grant Disclosures

The following grant information was disclosed by the authors:

Alexander von Humboldt Foundation.

Deutsche Forschungsgemeinschaft: 1012/28-1.

Bildungsund Dokumentationszentrum Ostbayerische Erdgeschichte e.V.

Freunde der Bayerischen Staatssammlung für Paläontologie und Geologie e.V.

## Competing Interests

The authors declare that they have no competing interests.

## Author Contributions

- Andrea Villa conceived and designed the experiments, performed the experiments, analyzed the data, prepared figures and/or tables, authored or reviewed drafts of the paper, and approved the final draft.
- Roel Montie performed the experiments, analyzed the data, authored or reviewed drafts of the paper, and approved the final draft.
- Martin Röper conceived and designed the experiments, authored or reviewed drafts of the paper, and approved the final draft.
- Monika Rothgaenger conceived and designed the experiments, authored or reviewed drafts of the paper, and approved the final draft.
- Oliver W.M. Rauhut conceived and designed the experiments, performed the experiments, analyzed the data, authored or reviewed drafts of the paper, and approved the final draft.

## Data Availability

The specimen studied is part of the collections of the Bayerische Staatssammlung für Paläontologie und Geologie (Munich, Germany): SNSB-BSPG 1993 XVIII 4.

## New Species Registration

The following information was supplied regarding the registration of a newly described species:

Publication LSID: urn:lsid:zoobank.org:pub:177F78D8-2C99-4C3B-8ED5-8D8ADE960A57.

*Sphenofontis* LSID: urn:lsid:zoobank.org:act:99B5062E-2A23-4221-A8FD-15329587A83E.

*Sphenofontis velserae* LSID: urn:lsid:zoobank.org:act:1224EDEE-668A-4766-A69D-E2F4E4C87890.

## Supplemental Information

Supplemental information for this article can be found online at http://dx.doi.org/10.7717/peerj.11363#supplemental-information.

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
