# Peer review of "Sphenofontis velserae gen. et sp. nov., a new rhynchocephalian from the Late Jurassic of Brunn (Solnhofen Archipelago, southern Germany)"

_PeerJ, doi:10.7717/peerj.11363_

## Round 0.1 · original submission · Minor Revisions

Dear authors,

I have accepted the reviewers' recommendation of ‘minor revisions’.

I look forward to receiving your revised manuscript.

·

Basic reporting

The structure of the manuscript is well organized and presented in clear and technically correct English. Figures, illustrations and captions have a good quality and are easy to understand. Bibliography is extensive and sufficient to provide a good background for the introduction and discussion of the work.

Experimental design

As mentioned above, the manuscript is well organized; however, it was frustrating that the authors did not include a phylogenetic analysis considering that their specimen is very complete and allows observing many taxonomic features worth to be included in a data matrix. If they were describing a specimen that only consist in fragments of dentary or maxilla, I could understand their decision to not to perform phylogenetic analysis, but this is not the case. I suggest including a parsimony analysis in order to improve their work.

Validity of the findings

After reviewing the manuscript, I found the results and conclusions of this work convincing and interesting enough to be considered for publication in PeerJ. I think the authors provide a well written and very detailed description of a fascinating specimen that provides remarkable information about some skeletal elements poorly studied in fossil rhynchocephalians as is the case of the pectoral and pelvic girdle. As mentioned before, the description of the new taxon is very detailed; but I felt that the diagnosis presented is somewhat plain. Therefore, in order to improve their work I suggest changing the current diagnosis for a differential diagnosis.

For example:
Sphenofontis velserae is moderated sized taxon that differs from other sphenodontians by the following unique combination of features. Proximally-constricted and strongly distally-expanded transverse processes of the first sacral vertebra (not present in any other fossil rhynchocephalian or in the extant Sphenodon); anterolaterally-oriented transverse processes of the first caudal vertebra (a similar transverse process is present in Sphenodon but ………).

Regarding to the taxonomic validity of the new taxon, I would like to make it clear that a) I have no doubt that the specimen represents a new genus and species, b) that this new taxon confirms the high taxonomic diversity of the Late Jurassic sphenodontians, and c) that Sphenofontis can be accurately referred to Neosphenodontia, and it likely can be placed within the Sphenodontinae. However, I was surprised with the fact that the authors did not include a phylogenetic analysis to provide more support to their conclusions on the taxonomic placement of Sphenofontis. As far as I am concerned, the authors present a work of an overall great quality, however, I suggest that before the final acceptance of their work, to include at least a basic parsimony analysis to test the phylogenetic position of the new taxon. If a phylogenic analysis is presented, I think is not necessary to do a further discussion on rhynchocephalian relationships, it is fine if they only point out if the results confirm the placement of Sphenofontis within the Neosphenodontia and the Sphenodontinae. An additional figure with a phylogenetic tree can be incorporated in the main manuscript, but it is also fine if the authors include it as part of the supplementary material.

Other comments and suggestions to the authors are as follow:

The authors cite several times the work of Romo-de-Vivar-Martinez et al., in press; nevertheless, that work was already published last year. Please, correct that in the text and in the references.

• Romo-de-Vivar-Martínez, P. R., Martinelli, A. G., Paes Neto, V. D., Scartezini, C. A., Lacerda, M. B., Rodrigues, C. N., & Soares, M. B. (2019). New rhynchocephalian specimen in the Late Triassic of southern Brazil and comments on the palatine bone of Brazilian rhynchocephalians. Historical Biology, 1-9.

Simões and colleagues recently published a review of sphenodontian phylogeny which also discusses a little bit about mosaic evolution in Sphenodon. I suggest the authors take a look of that work.

• Simões, T. R., Caldwell, M. W., & Pierce, S. E. (2020). Sphenodontian phylogeny and the impact of model choice in Bayesian morphological clock estimates of divergence times and evolutionary rates. BMC biology, 18(1), 1-30.

Additional comments

This is a very interesting study of a specimen that I considered fascinating when some pictures were published in a previous work. I am glad to see this specimen finally described, but I recommend taking into account my suggestions in order to improve your work and give more support to your findings.

·

Basic reporting

'no comment

Experimental design

'no comment

Validity of the findings

'no comment

Additional comments

The manuscript is well written, however, take into consideration that English is not my native language. The description is good, and I think is necessary continue producing this type of papers. Although when using the terms "Slender", "Robust", "longer", "Short" without a reference either in relation to the proportions of the bone element in question, or between said element in relation to other bone elements, it becomes quite The meaning of these terms is subjective and I think we should stop with this. Throughout the text the authors make mention of some proportions, a table could be attached that contains them and they are used as a reference in the descriptions of the bone elements.

Although, a phylogenetic analysis is not necessary to describe a new species, and even less with a detailed description and good comparison like the one presented in this work (and finally the authors mention that it will be done in the future). It seems to me that for the first part of the discussion it would help a lot, and in this case it would be necessary, to be able to effectively test the hypothesis about the phylogenetic location, mainly into the Sphenodontidae and to locate how the new taxon described relates to the other taxa. It would also help to map some characteristics that have been highlighted throughout the text and this would contribute to the understanding of the relationships and evolution of the group. I recommend checking out the new article: Simões, T. R., Caldwell, M. W., & Pierce, S. E. (2020). Sphenodontian phylogeny and the impact of model choice in Bayesian morphological clock estimates of divergence times and evolutionary rates. BMC biology, 18(1), 1-30.

I think it is important to improve some figures or to append some made with the help of a stereoscopic microscope since some structures mentioned in the text are not visible or are difficult to find. Or even in the interpretive drawings to point out some structures such as foramina, the autotomy plane, notch, ridges, etc. since sometimes it is difficult to see them or I do not know they see them in the photographs.

Finally, they could append one more figure to accompany the Intruduction sections, and Geological and Palentological Context, this figure should be a stratigraphic column (or several columns if they are the case and the relationship between them) where the mentioned fossils are located stratigraphically in the text, and wich they have been found in what is known as the Solnhofen Archipelago.

Finally, in the manuscript, attach some comments.

·

Basic reporting

This manuscript represents a detailed description of a new taxon from the Solnhofen Archipelago. The authors are to be congratulated on a very thorough job and an excellent comparison with the known record of Early Mesozoic rhynchocephalians. It will be a valuable contribution to our understanding of the increasingly complex radiation of the sister group of lizards.

There are very few things that need to be considered in the revision, although I would like to see at least some discussion of its potential habit – such a beautifully preserved specimen does at least require comments on the limb and body proportions when compared with other members of the group.

Experimental design

The approach is excellent and the figures clear and well-composed.

I am assuming that it is not possible to CT scan the specimen? You might like to comment on this as it would have potentially offer a solution for better understanding of the dorsal cranial morphology.

Validity of the findings

For the diagnosis of the new taxon, I completely agree that the “waisting” of the sacral rib on the first sacral is probably quite unique. However, I wonder if the small medially placed tooth on the maxilla might be aberrant? After all we are only dealing with one individual.

I have a handful of questions about your identification and comments on different elements:

L. 164. Difficult to interpret but I guess I have to take your word that you have identified the frontal correctly. It just looks a bit like the anterior ramus of the pterygoid and its position would make sense in this context.

L. 299. Is this not the medial view of the right quadrate so that what is labelled as the qj is the internal (columnar) part of the quadrate. The elements of the right side, although somewhat disarticulated are surely preserved in medial view as you indicate in L. 278?

L. 571. Additional teeth on the two maxillae seem to be different on either side. On the left it is the third successional tooth that is reduced and positioned more medial, whereas on the right side it is indeed the fourth successional tooth that is small and medial in position.

L.905. The femora seem relatively short to me. I guess it depends upon your point of reference - but when compared to the forelimbs. It would be worth comparing the proportions of the forelimbs and hindlimbs to other rhynchocephalians. Likewise the pedal digits seem short. I could be incorrect, but comparison with the forelimb:hindlimb ratios in other taxa/ specimens would be instructive.

L. 1036. I tend to agree that the dentition is unique and indicative of a new taxon, but should perhaps add the caveat that this is based on just one (not completely mature) individual, and that some variation in dentition might be expected.

L.1164. This is the only real comment that hints at further studies on functional morphology. As suggested above, it would be helpful if the authors could at least make some comments on limb proportions in the new taxon in this respect.

Additional comments

For the most part the text reads fairly well, but I have made a few suggested changes to phrasing in the attached PDF where I have felt there is either some ambiguity or the expression is particularly clumsy.

---

## Round 0.2 · accepted · Accept

Dear authors,

Thank you for your revised version of the manuscript. Looking over your response to the reviewers' comments I have decided to ‘accept’ your manuscript for publication.

In short order, the production staff will be in contact with you to take you through the proofing stages.

Thank you again for choosing PeerJ as your publication venue, and I hope you will use us again in the future.